# SMN promotes mitochondrial metabolic maturation during myogenesis by regulating the MYOD-miRNA axis

Akihiro Ikenaka[1], Yohko Kitagawa[1], Michiko Yoshida[2], Chuang-Yu Lin[1,3], Akira Niwa[1] , Tatsutoshi Nakahata[4], Megumu K Saito[1]

**Spinal muscular atrophy (SMA) is a congenital neuromuscular disease caused by the mutation or deletion of the *survival motor neuron 1 (SMN1)* gene. Although the primary cause of progressive muscle atrophy in SMA has classically been considered the degeneration of motor neurons, recent studies have indicated a skeletal muscle–specific pathological phenotype such as impaired mitochondrial function and enhanced cell death. Here, we found that the down-regulation of SMN causes mitochondrial dysfunction and subsequent cell death in in vitro models of skeletal myogenesis with both a murine C2C12 cell line and human induced pluripotent stem cells. During myogenesis, SMN binds to the upstream genomic regions of MYOD1 and microRNA (miR)-1 and miR-206. Accordingly, the loss of SMN down-regulates these miRs, whereas supplementation of the miRs recovers the mitochondrial function, cell survival, and myotube formation of SMN-deficient C2C12, indicating the SMN-miR axis is essential for myogenic metabolic maturation. In addition, the introduction of the miRs into ex vivo muscle stem cells derived from Δ7-SMA mice caused myotube formation and muscle contraction. In conclusion, our data revealed novel transcriptional roles of SMN during myogenesis, providing an alternative muscle-oriented therapeutic strategy for SMA patients.**

## Introduction

Spinal muscular atrophy (SMA) is an inherent neuromuscular disease caused by mutation or deletion of the *survival motor neuron 1* (*SMN1*) gene. *SMN1* encodes the SMN protein. In the severest form of SMA, infants suffer from severe muscle weakness and respiratory failure during the first year of life. Classically, reduced SMN expression was thought to cause selective and primary lower motor neuronal death, leading to subsequent denervation and muscle atrophy, because the degeneration of anterior horn motor

neurons is the predominant pathological finding of SMA. However, SMA is currently regarded as a systemic disorder affecting not only motor neurons, but also neuromuscular junctions (NMJs) and skeletal muscles (Hayhurst et al, 2012; Martinez et al, 2012). Indeed, myoblasts with SMN knockdown show reduced proliferation and fusion defects (Shafey et al, 2005). Animal models of SMA also revealed skeletal muscle has its own pathological contribution to the SMA phenotype (Martinez et al, 2012; Kim et al, 2020). However, the cell-autonomous molecular mechanism of the skeletal muscle degeneration because of SMN deficiency is mostly unknown.

SMN localizes ubiquitously and exerts its function in various manners. The first identified function of SMN was the assembly of spliceosomal small nuclear ribonucleoproteins (snRNPs). SMN forms a large protein complex (SMN complex) and chaperones the biogenesis of snRNPs in the cytoplasm and subsequent translocation to the nucleus (Qing Liu, 1997). In neurons, SMN has a unique role in mRNA transport and local translation, which is exerted by binding with HuD in the cytoplasm (Akten et al, 2011; Hao le et al, 2017). SMN is also involved in actin dynamics and axon elongation by interacting with profilin 2a (Sharma et al, 2005; Akten et al, 2011; Hao le et al, 2017). In addition, the loss of SMN in motor neurons can cause mitochondrial dysfunction and impaired cellular respiration (Miller et al, 2016). These findings indicate that SMN has both universal and cell type–specific roles. Thus, understanding the specific function of SMN in skeletal muscle could lead to novel muscle-targeting therapeutic strategies.

Skeletal muscle stem cells (MuSCs) and myoblasts undergo a dramatic bioenergetic transition from glycolysis to oxidative phosphorylation during differentiation into myotubes. This metabolic maturation is accompanied by mitochondrial maturation (Remels et al, 2010; Miller et al, 2016). One of the most important transcription factors governing the metabolic and mitochondrial maturation is *myoblast determination protein 1* (*MYOD1*). MYOD1 is known to regulate oxidative metabolism by directly binding the enhancers along oxidative metabolic genes (Shintaku et al, 2016). MYOD1 also contributes to metabolic maturation by up-regulating microRNA (miR)-1, miR-133a, and miR-206, all of which are

[1]Department of Clinical Application, Center for iPS Cell Research and Application, Kyoto University, Kyoto, Japan   [2]Department of Pediatrics, Kyoto Prefectural University of Medicine, Kyoto, Japan   [3]Department of Biomedical Science and Environmental Biology, Kaohsiung Medical University, Kaohsiung, Taiwan   [4]Drug Discovery Technology Development Office, Center for iPS Cell Research and Application, Kyoto University, Kyoto, Japan

Correspondence: msaito@cira.kyoto-u.ac.jp

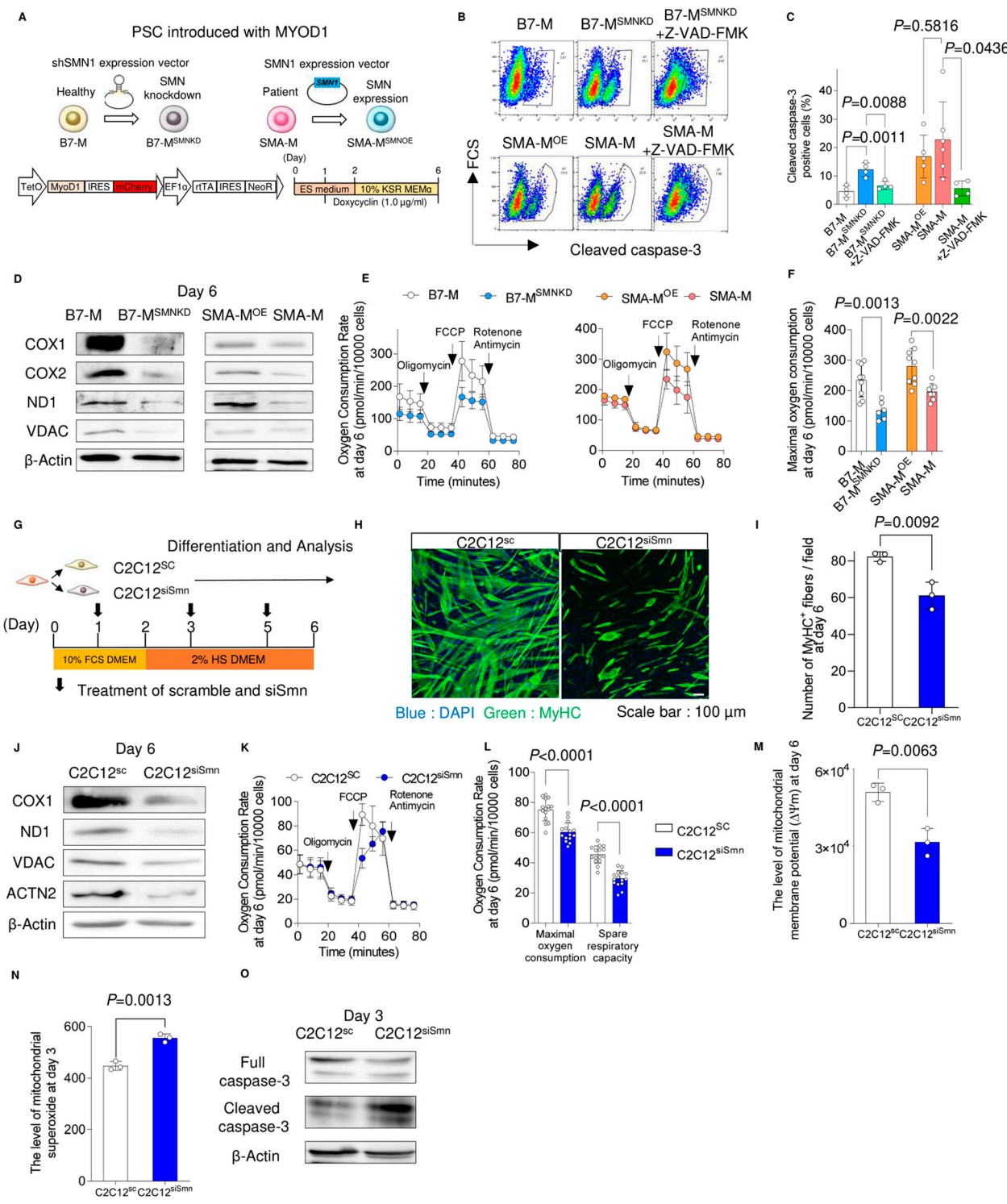

**Figure 1. Loss of SMN causes mitochondrial bioenergetic failure during myogenic differentiation.**
**(A)** Schema of iPSC clones used in the study and the doxycycline-inducible MYOD1-driven myogenic conversion system. **(B, C)** Cleaved caspase-3–positive apoptotic cells in human iPSC-derived myogenic cells (day 3) analyzed by intracellular flow cytometry. Z-VAD-FMK (20 μM) was added. **(B, C)** Representative flow diagrams and (C) their quantification. **(D)** Immunoblotting assay with iPSC-derived myogenic cells (day 6). **(E)** OCR measured in iPSC-derived myogenic cells (day 6). Oligomycin (10 μM), FCCP (10 μM), and rotenone (1.0 μM) plus antimycin (1.0 μM) were sequentially added. Data were obtained from $1.0 \times 10^4$ cells. **(E, F)** Maximal oxygen consumption was calculated using the data in (E). **(G)** Schema of myogenic differentiation with C2C12 cells. **(H)** Representative immunostaining images of C2C12 cells with MyHC (myosin heavy chain) (day 6). **(I)** Average of the number of MyHC-positive fibers per field (day 6). **(J)** Immunoblotting assay with C2C12 cells (day 6). **(K)** OCR measured with C2C12 cells (day 6). Oligomycin (10 μM), FCCP (10 μM), and rotenone (1.0 μM) plus antimycin (1.0 μM) were sequentially added. **(K, L)** Maximal OCR and spare respiratory capacity were calculated using the data in (K). **(K, L)** Value of oxygen consumption was normalized to $1.0 \times 10^4$ cells. **(M)** Mitochondrial membrane potential (Δψm) of C2C12 cells (day 6).

important for metabolic and mitochondrial maturation (Shintaku et al, 2016; Przanowska et al, 2020). The MYOD1-miR axis is therefore important for skeletal muscle biogenesis. Interestingly, skeletal muscle specimens obtained from SMA patients showed a down-regulation of electron transfer chain (ETC) and mitochondrial outer membrane proteins (Ripolone et al, 2015), a phenomenon also seen in neurons. Oxygen consumption was also decreased in an in vitro model using myotubes differentiated from SMA pluripotent stem cell (PSC) models (Ripolone et al, 2015; Hellbach et al, 2018). Although these findings highlight mitochondrial dysfunction in skeletal muscles, the precise molecular mechanism remains unknown. Especially, the relationship between SMN and the MYOD1-miR axis should be elucidated.

Here, we describe a novel function of SMN during myogenesis using human and murine in vitro models. We found that SMN diffusely localizes in the nucleus during myogenesis. The loss of SMN expression causes the down-regulation of MYOD1, miR-1, and miR-206, resulting in impaired metabolic maturation and myotube formation. We also show that the in vitro phenotypes of SMA myotubes are successfully rescued by the ectopic expression of miRs, demonstrating SMN acts as an upstream regulator of these genes. Interestingly, SMN binds to the upstream genomic regions of MYOD1 and the miR host genes, indicating that SMN regulates their expression. Our results highlight the unique stage- and cell type–specific functions of nuclear SMN that make it a regulatory factor for the metabolic maturation of myotubes, thus providing novel insights into the skeletal myogenesis and an alternative therapeutic strategy for the biogenesis of skeletal muscle in SMA.

# Results

### Loss of SMN causes mitochondrial bioenergetic failure during the myogenic conversion of iPSCs

Knowing that the forced expression of MYOD1 converts human PSCs to myotubes (Tanaka et al, 2013), we first investigated the effect of SMN down-regulation on the myogenic conversion of human iPSCs. For this purpose, we prepared two pairs of isogenic iPSC clones with a doxycycline-inducible MYOD1 expression construct: first, a control 201B7 iPSC line and its SMN-knockdown counterpart (B7-M and B7-M$^{SMNKD}$, respectively); and second, an SMA patient–derived iPSC line (Yoshida et al, 2015) and the same line but with SMN supplementation (SMA-M and SMA-M$^{OE}$, respectively). We then converted these clones into myogenic cells by adding doxycycline (Fig 1A). On day 3, more than 90% of the cells were positive for myogenin (MyoG) in all clones (Fig S1A and B). Genes associated with myogenesis, such as MyoG and myocyte enhancer factor 2C (MEF2C), were also up-regulated after differentiation (Fig S1C). The level of SMN protein was lower in B7-M$^{SMNKD}$ and SMA-M even after differentiation (Fig S1D). Interestingly, we found that although iPSC clones with sufficient SMN expression (B7-M and SMA-M$^{OE}$; SMN-maintained clones) showed increased cell number during the conversion, those

with down-regulated SMN (B7-M$^{SMNKD}$ and SMA-M; SMN-down-regulated clones) failed to proliferate (Fig S1E). This effect could be attributed to the increased apoptosis in the SMN-down-regulated clones, because the cleavage of caspase-3 was increased. Indeed, treatment with a pan-caspase inhibitor, Z-VAD-FMK, reduced the number of cleaved caspase-3–positive cells in SMN-down-regulated clones (Fig 1B and C).

During myogenic differentiation, the bioenergetic status shifts from glycolysis to mitochondrial oxidative phosphorylation. To enhance the mitochondrial function, genes associated with the mitochondrial respiratory complex are up-regulated (Remels et al, 2010; Wust et al, 2018). To understand the mechanism of apoptosis, we focused on the mitochondrial biology during myogenic conversion, because myogenic differentiation promotes mitochondrial biogenesis (Frangini et al, 2013) and because mitochondrial failure can cause apoptosis. During the myogenic conversion, the mitochondrial DNA copy number increased in SMN-maintained clones but not in SMN-down-regulated clones (Fig S1F). SMN-down-regulated myogenic cells showed a down-regulation of mitochondrial proteins associated with bioenergetic function (Fig 1D). The down-regulation of mitochondrial proteins generally represses the oxygen consumption rate of mitochondria (Trotta et al, 2017; Yan et al, 2019). Consistently, the mitochondrial oxygen consumption capacity was impaired (Figs 1E and F and S1G). In line with this finding, the mitochondrial membrane potential ($\Delta\psi m$) was lower in SMN-down-regulated clones (Fig S1H). These results indicate that SMN is required for mitochondrial bioenergetic maturation during myogenic conversion.

Impairment of the mitochondrial complex in the ETC promotes excessive ROS production, resulting in apoptosis (Trotta et al, 2017). Consistently, ROS production was increased in SMN-down-regulated clones on day 3 (Fig S1I). Treating the SMN-down-regulated clones with an antioxidant, $\alpha$-tocopherol, reduced the apoptotic cell number such that it almost equaled that of SMN-maintained clones (Fig S1J) and recovered the total cell number (Fig S1K). Overall, our observations suggest that the loss of SMN causes mitochondrial bioenergetic dysregulation and subsequent ROS-mediated apoptosis during myogenic conversion.

### Depletion of SMN causes mitochondrial dysfunction in C2C12 cells

Because the iPSC model uses an artificial expression of MYOD1 to convert cell fate, we next evaluated the reproducibility of our findings by employing a commonly used myogenic differentiation model with the murine C2C12 cell line. For this, we knocked down Smn in C2C12 (C2C12$^{siSmn}$) and differentiated the cells into myotubes by changing the culture medium (Fig 1G). C2C12$^{siSmn}$ showed reduced SMN expression at both the transcript and protein levels (Fig S1L and M). Differentiated C2C12$^{siSmn}$ showed decreased myotube formation, as measured by myosin heavy chain staining (Fig 1H and I), indicating the indispensable role of Smn in myogenic differentiation.

---

**(N)** Mitochondrial superoxide levels of C2C12 cells (day 3) evaluated with MitoSOX. **(O)** Immunoblotting assay with C2C12 cells (day 3). Error bars indicate means ± S.D. **(C)** Statistical analysis by one-way ANOVA with multiple comparisons. **(F, I, L, M)** Statistical analysis by a t test. Each dot represents a biologically independent sample.

▶▶▶▶ Life Science Alliance

The mitochondrial function of C2C12 cells was also evaluated. Cytochrome c oxidase subunit 1 (COX1) was up-regulated in C2C12 cells 6 d after the differentiation, indicating mitochondrial maturation (Fig S1N and O). On the contrary, mitochondrial proteins in differentiated C2C12^siSmn were less compared with C2C12^SC (Fig 1J). As expected, C2C12^siSmn showed a lower oxygen consumption capacity than C2C12^SC (Fig 1K and L), indicating the dysregulation of mitochondrial electron transmission. Consistently, Δψm was also decreased in C2C12^siSmn (Fig 1M).

As expected, mitochondrial ROS production increased in C2C12^siSmn on day 3 of the differentiation (Fig 1N). The level of cleaved caspase-3 also increased on day 3 in C2C12^siSmm, indicating the occurrence of mitochondrial apoptosis (Fig 1O). Overall, the data obtained with the C2C12 model were consistent with the iPSC model, showing reproducibility among different species and differentiation systems. Previous SMA studies about the pathogenesis of skeletal muscle showed a failure of the myogenic terminal differentiation, including the abnormal expression of myogenic regulatory genes and fewer myotubes (Nicole et al, 2003; Hayhurst et al, 2012; Boyer et al, 2013; Bricceno et al, 2014). Considering that the failure of the mitochondrial metabolic transition impairs myogenic terminal differentiation (Ripolone et al, 2015; Wust et al, 2018), our results indicate the potential relationship between the mitochondrial metabolic transition and failure of myogenic differentiation in SMA skeletal muscle models.

### Depletion of SMN down-regulates the expression of MYOD1 and downstream miR-1 and miR-206

MYOD1 is an essential transcriptional factor for myogenesis, and it and its downstream factors enhance mitochondrial oxidative metabolism during myogenic differentiation (Zhang et al, 2014; Shintaku et al, 2016; Wust et al, 2018). Because the myogenic differentiation of iPSCs, primary myoblasts, and C2C12 cells relies on the expression of MYOD1 (Tanaka et al, 2013; Wang et al, 2017), we hypothesized that SMN regulates muscle differentiation via MYOD1. In C2C12 cells, Myod1 expression increased during differentiation (Fig S2A), but the depletion of Smn significantly decreased the expression of Myod1 at both the mRNA and protein levels (Fig 2A and B). Interestingly, in the iPSC model, although the expression of an exogenous MYOD1 transgene was comparable between SMN-maintained clones and SMN-down-regulated clones (Fig S2B), the endogenous MYOD1 expression was significantly impaired in the SMN-down-regulated clones (Fig S2C). Therefore, the endogenous transcriptional control of the MYOD1 gene was affected by the dose of SMN.

We next investigated a downstream molecular mechanism harnessing MYOD1 and mitochondrial biogenesis. We focused on miR-1 and miR-206, because both are directly regulated by MYOD1, are highly expressed during the development of skeletal muscle (Rao et al, 2006), and control mitochondrial function in myogenic cells (Zhang et al, 2014; Wust et al, 2018; Przanowska et al, 2020). As expected, C2C12^siSmn showed less expression of miR-1 and miR-206 compared with C2C12^SC (Fig 2C and D). Similarly, in the iPSC model, SMN-down-regulated clones showed less expression too (Fig S2D and E).

Murine miR-1 and miR-206 are embedded in the miR1-1 host gene (miR1-1 hg) and long intergenic non-protein coding RNA muscle differentiation 1 (linc-MD1), respectively. These miRs are transcribed as primary miRs (pri-miRs) from these host genes. To determine whether the down-regulation of the miRs occurred at the transcriptional level, we evaluated the expression of their pri-miRs. The expressions of precursor miRs and the host genes of both miRs were also significantly down-regulated (Fig 2E and F), indicating that the cause of impaired miR expression can be at least partially attributed to transcriptional dysregulation. In conclusion, our data showed that the depletion of SMN reduces the expression of MYOD1 and its downstream miRs, which in turn could cause impaired mitochondrial metabolic maturation during muscle differentiation.

### Supplementation of miRs to SMN-depleted myoblasts recovers mitochondrial metabolism and myogenesis

To investigate whether miR-1 and miR-206 are responsible factors for the mitochondrial metabolic dysfunction in SMN-depleted myogenic cells, we introduced miR-1 and/or miR-206 into C2C12^siSmn before myogenic differentiation (Fig 3A). Supplementation of the miRs into C2C12^siSmn improved the oxygen consumption capacity on day 6 compared with untreated C2C12^siSmn (Fig 3B and C). Consistently, miR treatment recovered the expression of COX1 and ND1 to the level of C2C12^SC (Fig 3D–F). The effect of miR supplementation on mitochondrial metabolism was specific to C2C12^siSmn, as miR treatment to C2C12^SC did not affect the mitochondrial oxygen consumption capacity or the expression of COX1 (Fig S3A–C). Notably, miR supplementation improved the myogenesis of C2C12^siSmn, as it recovered the myotube formation ability and cell number (Fig 3G–I). In conclusion, the dysregulation of mitochondrial metabolism and myogenesis of myogenic cells because of SMN depletion is caused by the down-regulation of miRs.

Next, to address the causal association between the expression of MYOD1 and miRs, exogenous MYOD1 was introduced into C2C12 cells (Fig S3D). MYOD1 overexpression in C2C12^siSmn restored both the pri-miR and the precursor miR expression to levels comparable with C2C12^SC (Fig S3E and F). It also recovered the expression of proteins associated with mitochondrial metabolism (Fig S3G) and myotube formation (Fig S3H and I). Hence, MYOD1 depletion is responsible for the decrease of these miRs. Overall, our data demonstrate that SMN contributes to the maturation of mitochondrial metabolism and myogenesis by regulating the MYOD1-miR-1/miR-206 axis.

### SMN transiently localizes in the nucleus during myogenesis

We next sought to understand the molecular mechanisms by which SMN regulates the MYOD1-miR-1/miR-206 axis. For this purpose, we first tracked the expression and localization of SMN. In the iPSC model, SMN was up-regulated at both the transcript and protein levels (Fig 4A–C), peaking at day 3 of the conversion. Interestingly, we found that SMN protein diffusely localized in the whole nucleus, corresponding to the up-regulation of SMN (Fig 4D and E). The temporal localization of SMN in the nucleus was confirmed by protein extraction of the nuclear compartment (Fig 4F). Similar to the iPSC model, the up-regulation and diffuse nuclear localization of SMN were observed during the myogenic differentiation of C2C12 cells, which is different from the typical localization of SMN on Cajal

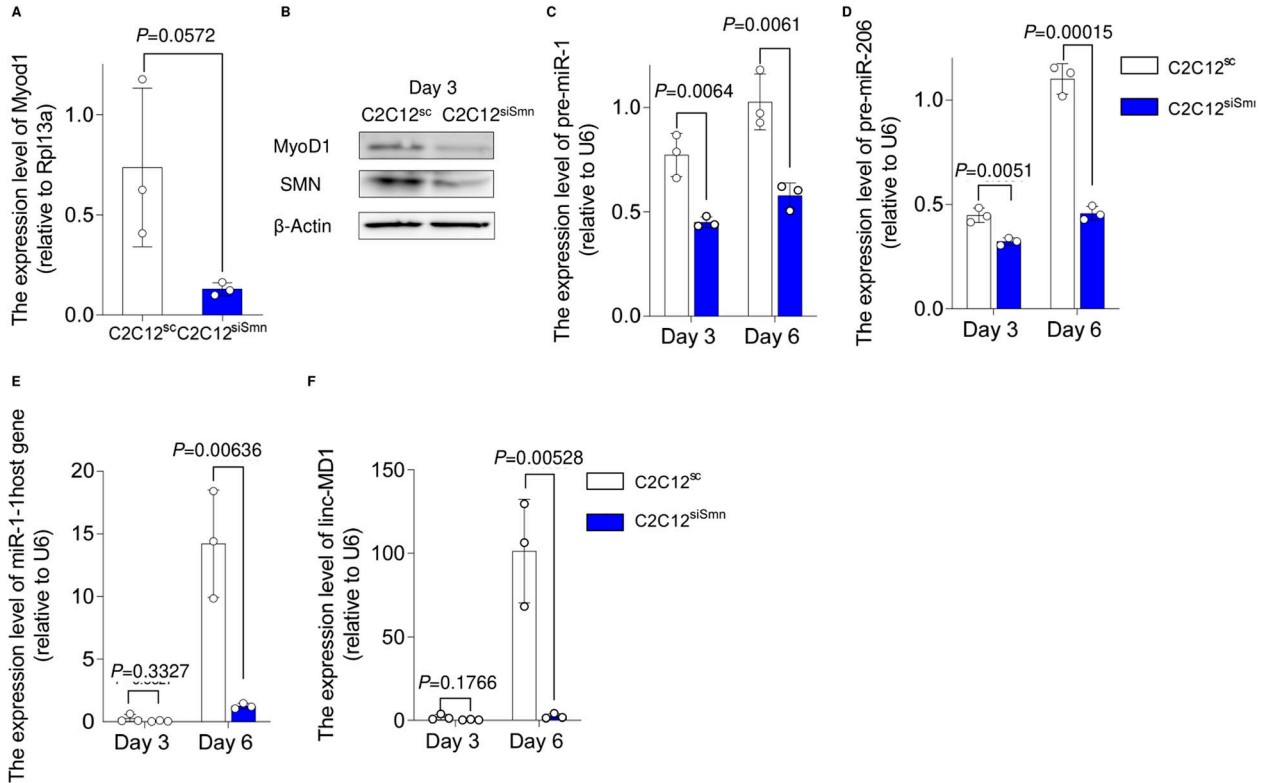

**Figure 2. Depletion of SMN down-regulates the expression of MYOD1 and its downstream targets miR-1 and miR-206.**
**(A, B)** Expression of Myod1 in C2C12SC and C2C12siSmn (day 6) at the (A) mRNA and (B) protein level. **(A)** Rpl13a served as the internal control. **(B)** β-Actin was used as the loading control. **(C, D, E, F)** qRT-PCR analysis for the expression of (C) pre-miR-1, (D) pre-miR-206, (E) miR1-1 host gene, and (F) linc-MD1. U6 served as the internal control. Error bars indicate means ± SD. Statistical analysis by a *t* test. Each dot represents a biologically independent sample.

bodies (Fig S4A–D). Therefore, we hypothesized that during myogenesis, SMN may have a unique role in the nucleus.

The canonical function of SMN is to form snRNPs in the cytoplasm and nuclear Cajal bodies in order to maintain the spliceosome (Zhang et al, 2008; Lotti et al, 2012). The number of nuclear SMN foci, which represents the localization of SMN in Cajal bodies, transiently increased in iPSC-derived myogenic cells (Fig S4E). However, the signal intensity of diffusely localized SMN protein was not significantly different between cells with and without foci (Fig S4F). Furthermore, SMN foci in C2C12 cells did not increase after differentiation despite the nuclear translocation of SMN (Fig S4G). These findings imply that diffusely distributed nuclear SMN is regulated independently of SMN in foci and that this spatiotemporally specific distribution is associated with the function of SMN during myogenesis.

### SMN and MYOD1 interact with each other and bind to the promoter regions of MYOD1, miR 1-1 hg, and linc-MD1

Previous studies reported that SMN is involved in genome instability and transcriptional termination by binding with RNAPLII (RNA polymerase II) (Zhao et al, 2016; Jangi et al, 2017). However, it is unclear whether SMN interacts with other molecules involved in transcription or whether it has a regulatory role in controlling cell fate by interacting with the genome. The impaired expression of endogenous MYOD1 mRNA in SMN-depleted cells and the temporal

diffuse nuclear localization of SMN during myogenesis (Figs 2A, 4E, and S2C) prompted us to test the hypothesis that the autoregulation of MYOD1 promoter is controlled by SMN. Indeed, chromatin immunoprecipitation–quantitative PCR analysis (ChIP–qRT-PCR) revealed that SMN bound to the upstream region of the transcription start site (TSS) of MYOD1 in both C2C12- and iPSC-derived myogenic cells (Figs 5A and S5A). Furthermore, we found that SMN and MYOD1 were co-immunoprecipitated with each other in both cell types (Figs 5B and S5B). A physical interaction between SMN and MYOD1 was also confirmed by the overexpression of Flag-tagged MYOD1 and His-tagged SMN in HEK293 cells (Fig S5C). Overall, our data suggest that SMN binds to the promoter region of MYOD1 interacting with MYOD1 and regulates the expression of MYOD1 during myogenesis.

Because the transcription of pri-miR-1 and pri-miR-206 is regulated by MYOD1 (Rao et al, 2006; Cesana et al, 2011) and because SMN and MYOD1 interact at the promoter region of MYOD1, we next investigated whether SMN also binds to the promoter region of the host genes of miRs. linc-MD1 has two promoter regions, a distal promoter and proximal promoter, in its locus (Cesana et al, 2011). The binding of SMN in both promoter regions was confirmed by ChIP–qRT-PCR analysis (Fig 5C). SMN also bound to the promoter region of miR1-1 hg (Fig 5D). In addition, we confirmed that SMN binds to the promoter region of human miR-1 and miR-206 in 201B7-derived myogenic cells (Fig S5D and E). Taken together, SMN binds to the promoter regions of MYOD1 and miR host genes during

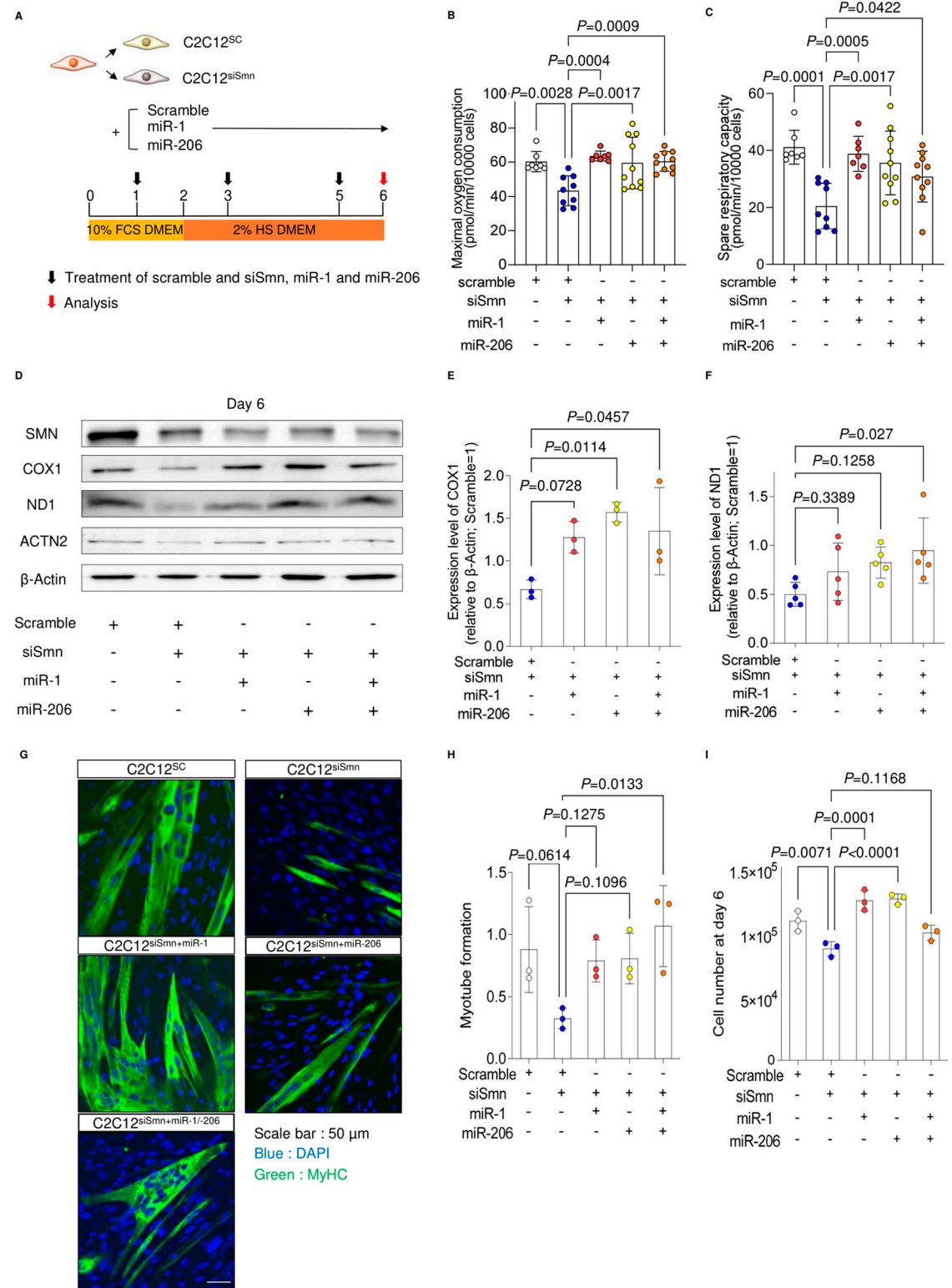

**Figure 3. Supplementation of miRNA recovers the phenotypes of SMN-depleted C2C12 cells.**
**(A)** Schema of the culture system. **(B, C)** (B) Maximal oxygen consumption and (C) spare respiratory capacity of C2C12siSmn cells (day 6). **(B, C)** Value of oxygen consumption was normalized to $1.0 \times 10^4$ cells, after the measurement of oxygen consumption. **(D, E, F)** (D) Immunoblotting assay with C2C12 cells (day 6), and (E, F) quantification of COX1 and ND1 protein. $\beta$-Actin served as the loading control. Relative values to C2C12SC (day 6) are shown. **(G)** Representative immunostaining images of C2C12 cells (day 6). **(H)** Myotube formation was defined by the ratio of the DAPI-positive area to the MyHC-positive area (day 6). **(I)** Number of C2C12 cells (day 6). Error bars indicate means ± SD. Statistical analysis by one-way ANOVA with multiple comparisons. Each dot represents a biologically independent sample.

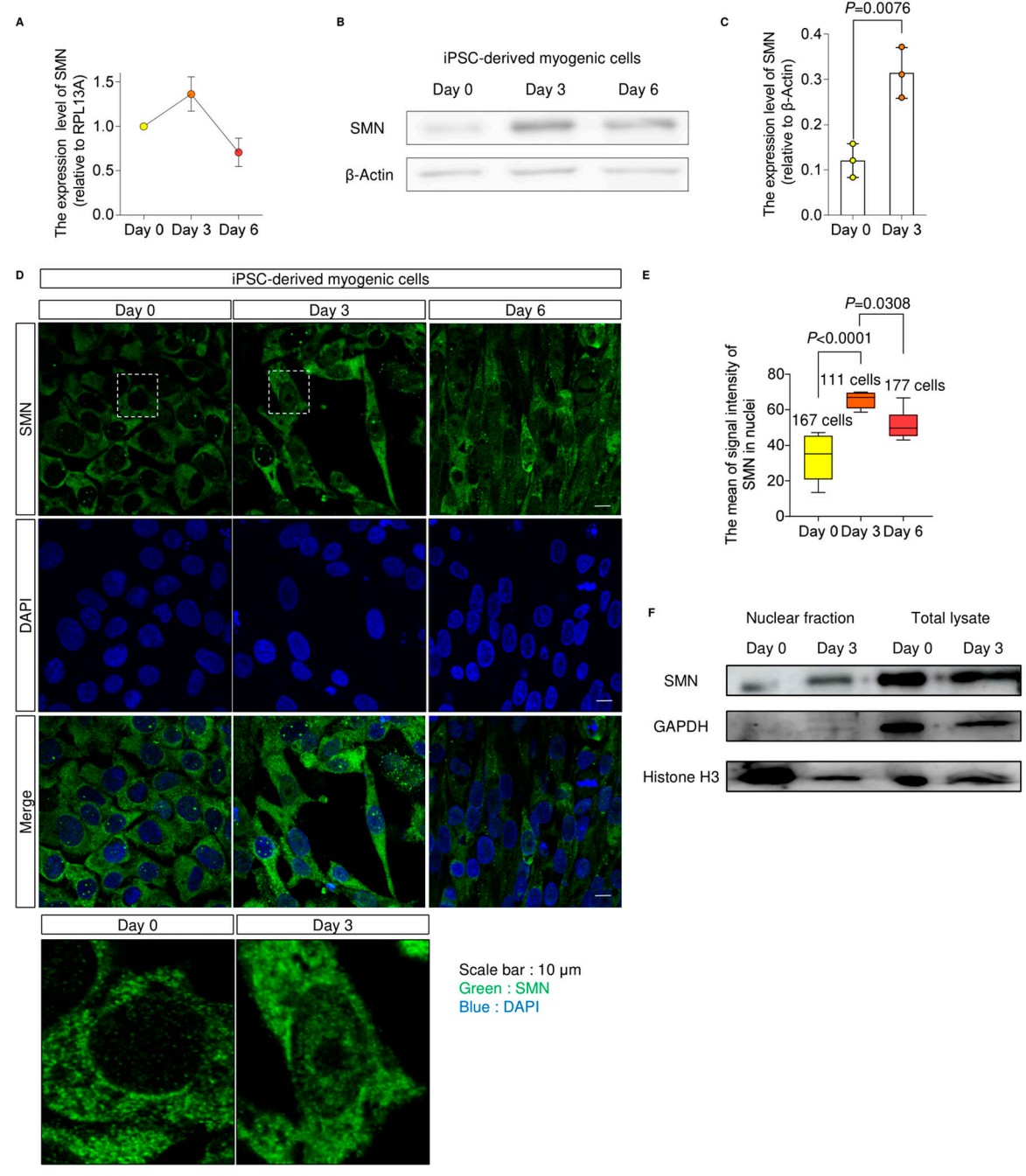

**Figure 4. SMN transiently localizes in the nucleus during myogenesis.**
**(A)** SMN expression in B7-M–derived myogenic cells. Relative values to the expression level at day 0 are plotted. RPL13A was used as the internal control.
**(B)** Immunoblotting assay with B7-M–derived myogenic cells. β-Actin served as the loading control. **(B, C)** Quantification of the SMN expression measured from (B).
**(D)** Representative immunostaining images of 201B7 iPSC- and B7-M–derived myogenic cells. Magnified images of the dotted square regions are shown below.
**(D, E)** Quantification of the signal intensity of nuclear SMN using the data in (D). **(F)** Immunoblotting assay with the nuclear fraction separated from 201B7 iPSC- and B7-M–derived myogenic cells. GAPDH and Histone H3 served as the loading control for total lysate and nuclear fraction, respectively. Error bars indicate means ± SD.
**(C)** Statistical analysis by a *t* test. **(E)** Statistical analysis by one-way ANOVA with multiple comparisons. Each dot represents a biologically independent sample.

myogenesis, suggesting that SMN may be involved in the transcriptional regulation of these genes.

Although these findings indicate the potential involvement of SMN in the transcriptional regulation of these genes, because SMN also binds to RNAPLII (Zhao et al, 2016), these results do not

exclude the possibility that SMN, along with RNAPLII, non-specifically binds the loci of actively transcribed genes. To test the specificity of the binding of SMN, we examined the binding of SMN on the promoter regions of another Myod1-driven gene *Myog* and a ubiquitously expressed gene *ribosomal*

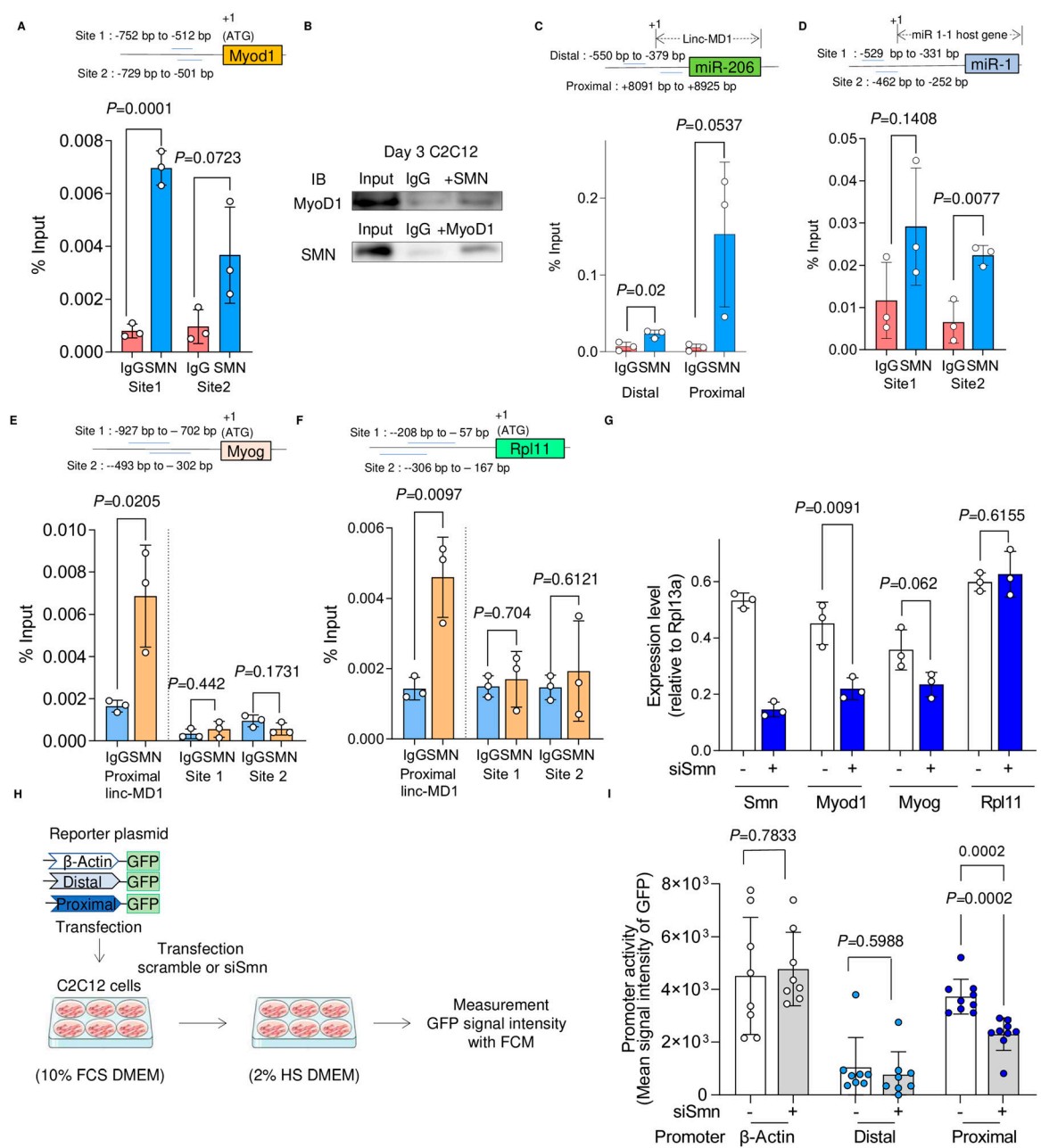

**Figure 5. SMN binds the upstream region of MyoD1 and miRs in C2C12-derived myogenic cells.**
**(A)** ChIP–qRT-PCR analysis of the SMN binding sites upstream of the MyoD1 TSS. C2C12 cells (day 3) were subjected to the analysis. The first ATG codon sequence was defined as +1. Blue lines indicate the target sites. **(B)** Co-immunoprecipitation assay of MyoD1 and SMN with C2C12 cells (day 3). **(C, D)** ChIP–qRT-PCR analysis of the SMN binding sites upstream of (C) miR-206 and (D) miR-1 in C2C12 cells (day 3). Blue lines indicate the target sites. The first base of the coding sequence of linc-MD1 or miR 1-1 hg was defined as +1. **(E, F)** ChIP–qRT-PCR analysis upstream of linc-MD1: (E) Myog and (F) Rpl11. The binding of SMN in C2C12 cells (day 3) was evaluated. **(G)** RT–qRT-PCR analysis for the expression of the indicated genes in C2C12SC and C2C12siSmn (day 3). Rpl13a was used as the internal control. **(H, I)** Schema of promoter activity assay with C2C12 cells (H) and quantification of GFP signal intensity with flow cytometry (I). Error bars indicate means ± SD. Each dot represents a biologically independent sample. Statistical analysis by a *t* test.

*protein L11* (*Rpl11*), and found that SMN did not bind to the promoter region of *Myog* and *Rpl11* (Fig 5E and F). As expected, the expression of these genes was not significantly different between C2C12^siSmn and C2C12^sc (Fig 5G). This indicates that there is a unique specificity in the genomic binding region of SMN, which may affect gene expression.

## SMN directly regulates transcription in a linc-MD1 proximal promoter-dependent manner

The genomic binding of SMN suggests the involvement of this molecule in transcriptional regulation. However, SMN is also involved in a variety of post-transcriptional processes, and the

deletion of SMN may cause significant perturbations in the gene expression profile of cells. Therefore, we performed a reporter assay using defined distal and proximal promoters of linc-MD1 (Cesana et al, 2011) to assess the direct effect of SMN on transcriptional activity. We transfected C2C12 cells with reporter plasmids incorporating GFP sequences directly under the proximal or distal promoter sequences of linc-MD1 and measured GFP fluorescence intensity during myogenesis (Fig 5H). GFP under the β-actin promoter sequence (−585 to −1 bp, referring to the ATG initiation codon) was used as a positive control. In line with the ChIP–qRT-PCR analysis of linc-MD1, the depletion of SMN did not suppress the GFP fluorescence under the distal promoter region of linc-MD1 or the promoter region of β-actin, whereas the fluorescence intensity under the proximal promoter region of linc-MD1 was significantly suppressed in C2C12$^{siSmn}$ (Fig 5I). Thus, SMN indeed binds to the proximal promoter region of linc-MD1 and regulates transcription.

### ChIP-sequencing (ChIP-seq) analysis of SMN reveals specific binding of SMN to certain MYOD1-regulated genes

Thus far, the results suggest that SMN interacts with MYOD1 and specifically regulates transcription on the genome for some MYOD1-regulated factors. However, the genome-wide specificity of SMN binding regions and its overlapping with the MYOD1 binding region are not clear. To address this, we performed ChIP-seq analysis using iPSCs with a doxycycline-inducible FLAG-tagged MYOD1 for MYOD1 ChIP-seq. Focusing on the effect of SMN on transcription, we first classified all genes into three clusters according to the binding pattern of RNAPLII using our ChIP-seq analysis of RNAPLII (Fig S5F). Cluster 1 was comprised of RNAPLII-unbound genes, cluster 2 was comprised of those in transcribed state, and cluster 3 was comprised of those in a paused state (Fig S5F) (Shao & Zeitlinger, 2017). We focused the subsequent analysis on clusters 2 and 3, in which RNAPLII bound to the genes. MYOD1 and RNAPLII were preferentially localized in the TSS region, whereas SMN tended to bind around the both TSS and transcriptional end site regions (Fig S5G). Next, to visualize where SMN, MYOD1, and RNAPLII bind to the gene structures, the relative enrichment of colocalized regions in each gene structure type was analyzed (Fig S5H). The colocalization sites of SMN, MYOD1, and RNAPLII were found to be enriched in transcriptional end site and 3'-UTR (Fig S5H), most likely reflecting the functions of SMN in transcription termination (Zhao et al, 2016). We also observed colocalization enrichment in promoter regions (Fig S5H), indicative of SMN-mediated transcriptional regulation in promoter regions. Next, we investigated the overlapping genomic regions of SMN, MYOD1, and RNAPLII in the whole genome. SMN and RNAPLII were found to colocalize, but not with MYOD1, in 558 genomic regions (Fig S5I). SMN and RNAPLII bound in the promoter region of 20 genes located in the 553 genomic regions (Table S1). On the contrary, SMN, MYOD1, and RNAPLII colocalized in 131 genomic regions (Fig S5I), and these three factors bound to the promoter region of 16 genes, including linc-MD1 (Table S1). In addition, in the 75 genes in the genomic region, SMN, MYOD1, and RNAPLII cobound with the gene loci outside of the promoter region (Table S1). SMN binding was not observed on some of the MYOD1-bound genes, such as *MYOG*, *MEF2*, and *MRF4*, nor on ubiquitously expressed genes, such as *RPL11*, *RPL13*, and *TUBB3* (data not shown). Collectively, the genome-wide analysis of the SMN binding

region by ChIP-seq revealed that SMN cooperates with RNAPLII and binds to the TSS region of certain MYOD1-regulated genes. Combined with the results of the reporter assay (Fig 5H and I), SMN is likely to be specifically involved in the transcriptional regulation of these genes.

### Loss of mitochondrial integrity occurs before denervation in skeletal muscle of SMA model mice

Several studies have suggested mitochondrial dysfunction in SMN-deficient cells, but none have described this event in the skeletal muscle of SMA model mice. We therefore wondered whether SMN and miR-1 and miR-206 have important roles in mitochondrial homeostasis in animal models. We used common SMA model mice, Δ7-SMA, for these experiments. Δ7-SMA mice are a transgenic strain expressing human SMN2 and the cDNA of SMNΔ7 in Smn-knockout mice (Smn$^{−/−}$; SMN2$^{+/+}$;SMNΔ7$^{+/+}$). They showed an extended lifespan of 9–18 d compared with the severest SMA model mice. To understand the skeletal muscle-specific pathology, we examined Δ7-SMA mice on postnatal day 3 (P3), which is before the occurrence of motor neuronal deficits. The body weight of P3 Δ7-SMA mice was already decreased compared with WT, as previously reported (Ando et al, 2020) (Fig S6A), indicating that skeletal muscle atrophy had already begun. The expression level of sarcomeric protein actinin Alpha 2 (ACTN2) also decreased in the tibialis anterior (TA), gastrocnemius (GA), and diaphragm of Δ7-SMA mice (Fig 6A), which is a hallmark of skeletal muscle atrophy (Schiaffino et al, 2013). Quantification of the mitochondrial area with transmission electron microscopy (TEM) images revealed smaller mitochondria in the TA and GA of Δ7-SMA mice (Figs 6B and C and S6B). In addition, SMN protein was hardly detected in the TA or GA of Δ7-SMA mice, and mitochondrial proteins were also fewer (Fig 6D and E). We confirmed that motor neurons and NMJs were maintained in P3 Δ7-SMA mice, because the areas of two neuronal markers (Tuj1 and SV2) and acetylcholine receptors were comparable to WT (Fig S6C–E). Therefore, the early mitochondrial defects in Δ7-SMA mice seem to be an intrinsic event of skeletal muscle preceding neuronal degeneration.

Severe SMA patients suffer from respiratory failure because of the paralysis of respiratory muscles (Burghes & Beattie, 2009). Considering that a loss of NMJs occurs before motor neuronal death in SMA model mice (Yoshida et al, 2015) and mitochondrial function is important for the maintenance of NMJs (Xiao et al, 2020), we speculated that postsynaptic mitochondria surrounding NMJs were also affected in the early life period. Postsynaptic mitochondria derived from the diaphragm of Δ7-SMA mice were significantly smaller than those from WT (Fig S6F and G). Mitochondrial proteins in the diaphragm of Δ7-SMA mice were also decreased (Fig S6H). Considering that the density of NMJs was maintained in the diaphragm of Δ7-SMA mice (see Fig S6I), these results suggest that the mitochondrial defects occur before the degeneration of NMJs.

### Impaired functional myotube-forming ability of MuSCs from Δ7-SMA mice is recovered by miR replacement therapy

Because in vivo myogenic processes are influenced by various environments, to evaluate the cell-autonomous properties of skeletal MuSCs derived from Δ7-SMA mice, CD45$^−$ CD31$^−$ Sca-1$^−$ Integrin α7$^+$ cells (Ieronimakis et al, 2010; Hayhurst et al, 2012) were sorted from skeletal muscle fibers and differentiated into myotubes

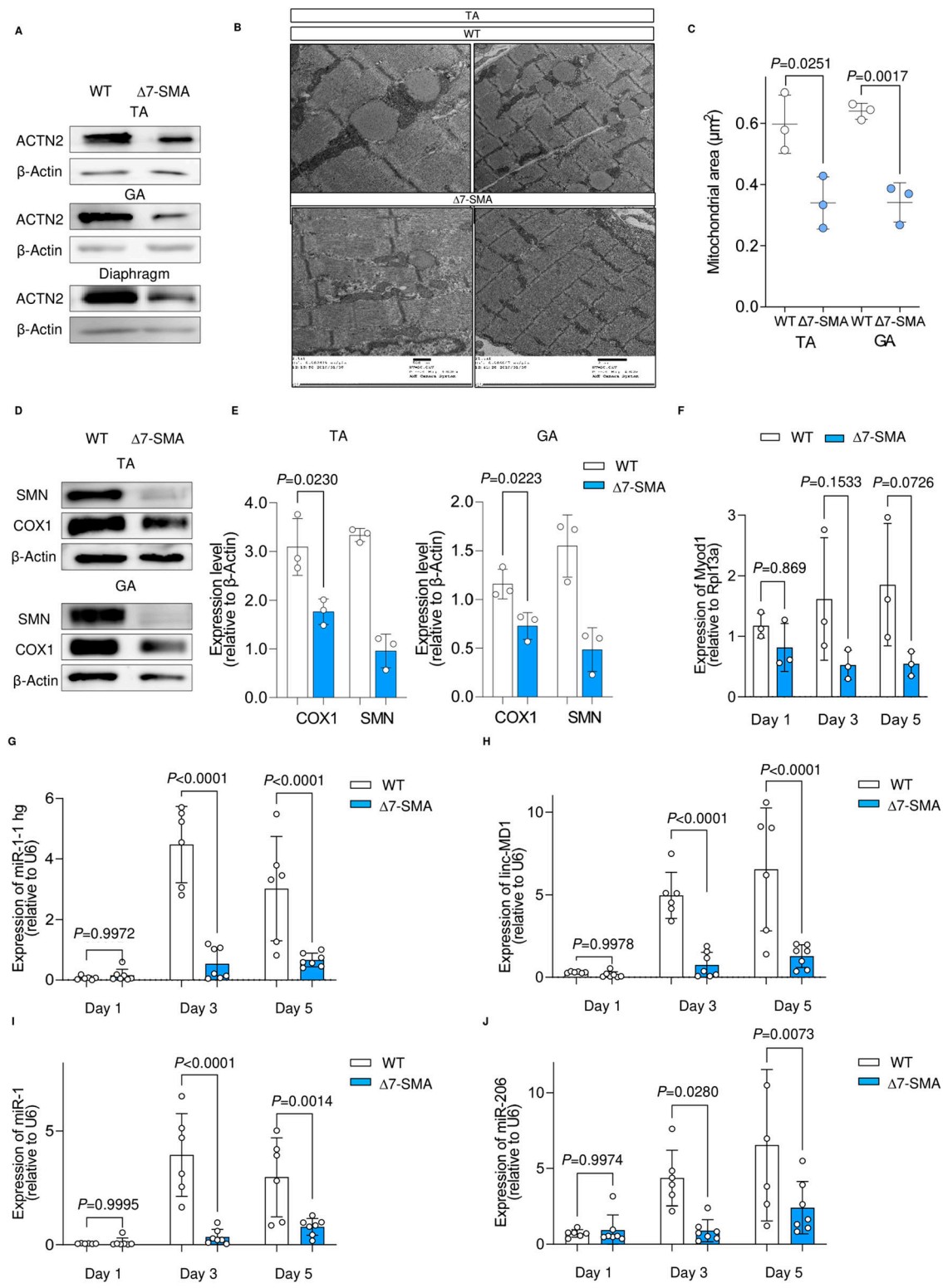

**Figure 6. Early muscular phenotype of neonatal Δ7-SMA mice.**
**(A)** Immunoblotting assay of TA, GA, and diaphragm samples from mice (P3). **(B)** Representative TEM images of TA muscle from WT mice and Δ7-SMA mice (P3).
**(C)** Quantification of the mitochondrial area (μm²) in TA and GA using data from the TEM images. Three mice from each group were evaluated. The number of analyzed mitochondria is as follows: TA (WT = 286, Δ7-SMA = 286) and GA (WT = 183, Δ7-SMA = 253). **(D, E)** (D) Immunoblotting assay with TA and GA samples from mice (P3) and (E) its quantification. **(F, G, H, I, J)** qRT-PCR analysis for the sequential expression of (F) Myod1, (G) miR-1-1 hg, (H) linc-MD1, (I) miR-1, and (J) miR-206 in myogenic cells derived from mice. Rpl13a and U6 were used as the internal control. Each dot represents independent mice. Error bars indicate means ± SD. **(C, E, F, G, H, I, J)** Statistical analysis by one-way ANOVA with multiple comparisons (F, G, H, I, J) and a *t* test (C, E). β-Actin served as the loading control.

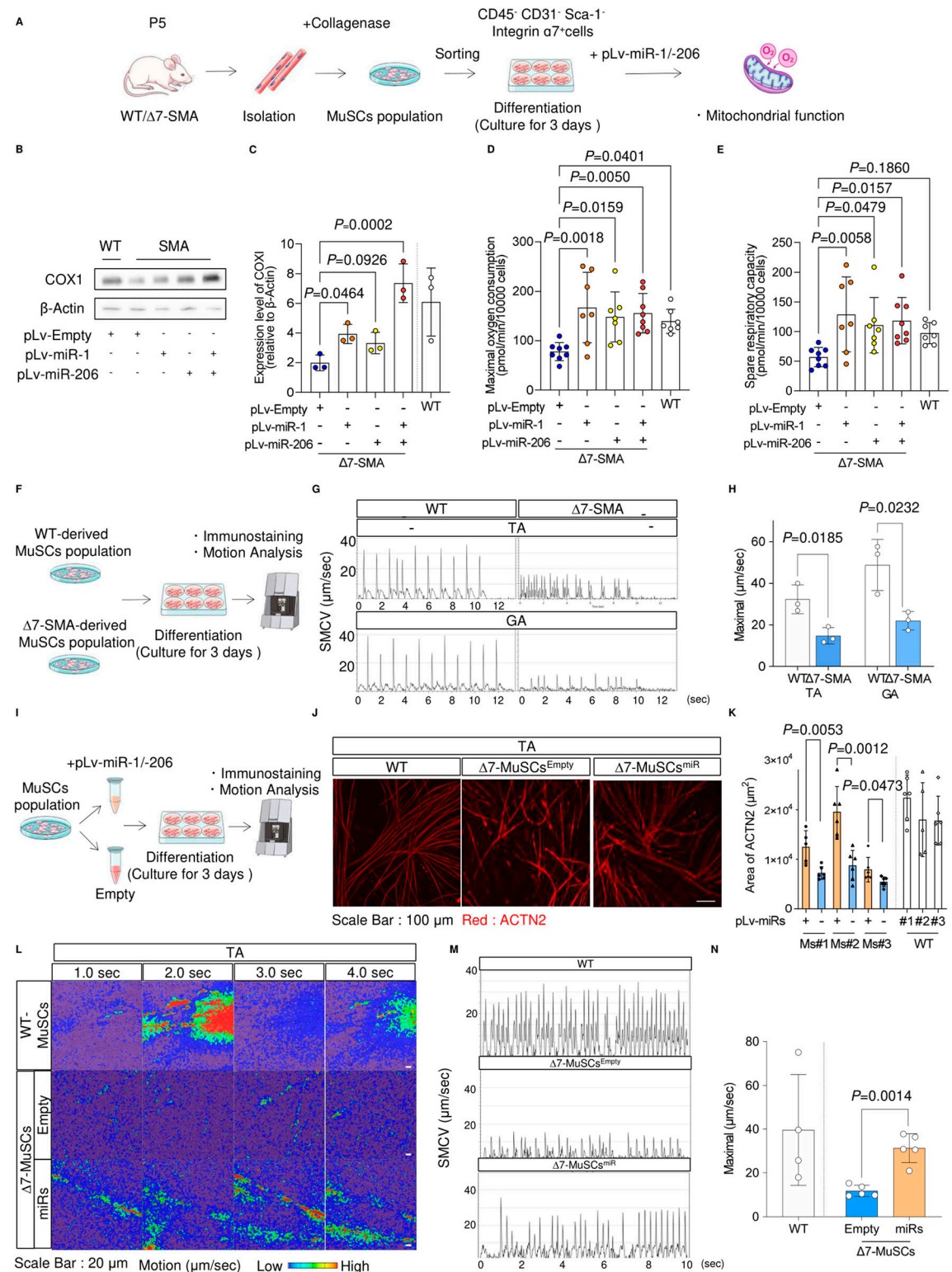

**Figure 7. miRNA treatment improves the ex vivo function of MuSC-derived myotubes from Δ7-SMA mice.**
**(A)** Procedure for the isolation of MuSCs, myotube differentiation, and analysis of mitochondrial function. **(B, C)** Immunoblotting assay of COX1 with limb samples from mice (P3) (B) and its quantification (C). **(D, E)** Maximal oxygen consumption (D) and spare respiratory capacity (E) of WT-MuSC–, Δ7-MuSC[Empty]–, and Δ7-MuSC[miRs]–derived myotubes. The value of oxygen consumption was normalized to 1 × 10[4] cells. **(F)** Procedure for motion analysis of MuSC-derived myotubes. SMCV, skeletal muscle contraction velocity. **(G)** Representative image of sequential SMCV of WT-MuSC– and Δ7-MuSC–derived myotubes. Top: MuSC-derived myotubes obtained from the TA. Bottom: MuSC-derived myotubes obtained from the GA. **(H)** Maximal SMCV. Three mice from each group were evaluated. The number of analyzed ROIs is as follows: TA

(Fig S6J). As expected, myogenic cells from MuSCs from Δ7-SMA mice (Δ7-MuSCs) down-regulated Myod1, miR-1, and miR-206 and their host genes (Fig 6F–J). Δ7-MuSC–derived myotubes also down-regulated COX1 and were associated with a lower maximal oxygen consumption rate (Fig 7B–D). Thus, myotubes from Δ7-SMA–derived MuSCs have reduced gene expression of the Myod1-miR axis, causing impaired mitochondrial function.

Previous data have shown that miR treatment alters the phenotype of skeletal muscle cell lines. Therefore, we next examined whether miRs could also improve the SMN-specific phenotype in mouse skeletal MuSCs. We applied miRNA replacement to MuSCs obtained from Δ7-SMA mice. To evaluate the potential therapeutic ability of miR supplementation, Δ7-MuSCs were infected by lentivirus containing either miR-1 or miR-206 expression vector or both (Δ7-MuSCs^miRs), or an empty vector (Δ7-MuSCs^Empty) (Fig 7A). The infection efficiency of miRs into MuSCs was high, and the introduced miRs were successfully up-regulated (Fig S7A and B). Similar to the C2C12 model, Δ7-MuSC^miRs–derived myotubes were found to restore COX1 expression (Fig 7B and C) and improved the mitochondrial oxygen consumption capacity (Fig 7D and E).

We next evaluated the contraction ability of the skeletal myotubes. For this, we stimulated them with a square-wave electric current and measured the skeletal muscle contraction velocity (SMCV) using a cell motion imaging system (Fig 7F) (Hoang et al, 2019; Lin et al, 2019). Myotubes from Δ7-MuSC showed a significantly slower SMCV than those from MuSCs derived from WT mice (WT-MuSCs) (Fig 7G and H). Consistent results were obtained for MuSCs from both the TA and the GA (Fig 7G and H). SMCV was not affected by treatment with curare, a competitive inhibitor of acetylcholine receptors, excluding the possibility of NMJ-dependent muscle contraction because of unexpected contamination of motor neurons (Fig S7C and D). Overall, Δ7-MuSCs had an impaired ability to differentiate into functional myotubes.

Finally, we evaluated whether miR supplementation in Δ7-MuSCs improved myotube formation and SMCV (Fig 7I). The introduction of the miRs significantly improved myotube formation compared with Δ7-MuSC^Empty–derived myotubes (Figs 7J and K and S7E and F). Notably, the SMCV of Δ7-MuSC^miRs-derived myotubes was significantly improved and reached the level of WT-MuSC–derived myotubes (Figs 7L–N and S7G–I and Video 1, Video 2, Video 3, Video 4, Video 5, Video 6, Video 7, Video 8, Video 9, Video 10, Video 11, and Video 12). Collectively, SMN plays an important role in functional muscle differentiation even in primary MuSCs, and miR replacement fully restored the differentiation potential of SMA-derived MuSCs.

## Discussion

Recently, mitochondrial metabolic dysfunction in SMA has been reported in both motor neurons and skeletal muscle (Ripolone et al, 2015; Miller et al, 2016). In the current study, we show that SMN deficiency caused mitochondrial metabolic dysfunction in C2C12 cells and myogenic cells derived from iPSCs at the early stage of differentiation. Moreover, Δ7-SMA mice showed few mitochondria and less mitochondrial protein expression at P3, which is before the occurrence of motor neuron denervation. Increasing mitochondrial ETC protein expression in C2C12siSmn- and Δ7-MuSC–derived myotubes by miR-1 and miR-206 treatment improved the myotube formation. Therefore, our results indicated a strong association of muscle atrophy in SMA with mitochondrial metabolic dysfunction during myogenic differentiation, indicating a possible pathway for the SMA pathology in skeletal muscle. It is difficult to strictly distinguish whether the decrease in miRs directly affects mitochondrial maturation or is a consequence of a reduced differentiation of myoblasts to myotubes. In the initial myogenic differentiation, miR-1 and miR-206 have an alternative target through which they promote differentiation in addition to mitochondria, and linc-MD1 also works as competing endogenous RNA in the cytoplasm (Chen et al, 2006, 2010; Cesana et al, 2011). Therefore, the down-regulation of SMN could impair both cytoplasmic and mitochondrial pathways during the initial myogenic differentiation. Mitochondrial maturation is an important event, but the coordinated differentiation program is essential to promote differentiation (Chen et al, 2006, 2010; Shintaku et al, 2016). In this study, we focused on the MyoD1-miR mitochondrial maturation pathway, and the contribution of other pathways to SMN muscle maturation will be investigated in future studies.

Our results indicated that SMN plays a role in the regulation of transcription during myogenic differentiation. However, there are still questions about this mechanism. SMN protein binds with RNAPLII via the carboxy-terminal domain to form a DNA-nascent RNA hybrid structure (R loop) (Zhao et al, 2016). Failure to resolve the R loop disturbs transcriptional termination and accelerates the accumulation of DNA damage (Zhao et al, 2016; Jangi et al, 2017; Grunseich et al, 2018). However, there is no evidence that SMN deficiency in the RNAPLII complex directly causes the down-regulation of transcripts. There are two hypotheses about the dysfunction of transcriptional regulation mediated by SMN on the promoter region. First is that SMN deficiency fails to form the R loop efficiently, leading to transcriptional down-regulation (Grunseich et al, 2018). This failure could alter the epigenetic modification of target genes, because the R loop structure in the promoter region disturbs the binding of epigenetic modifiers. Second is that SMN could form the transcription preinitiation complex, which is essential for transcription. Thus, SMN deficiency could disturb the formation of the preinitiation complex in the promoter region of target genes. Another possible factor for the tissue-specific phenotypes in SMA is the interaction between SMN and RNAPLII. Our results showed the specific binding of SMN on the promoter region of certain expressed genes. The expression of SMN-bound genes

---

(WT = 45, Δ7-SMA = 45) and GA (WT = 48, Δ7-SMA = 45). **(I)** Procedure for miRNA introduction into MuSCs. **(J)** Representative immunostaining images of MuSC-derived myotubes obtained from the TA. **(J, K)** ACTN2-positive area (μm²) was calculated from (J). Three mice from each group were evaluated. Each dot represents an ROI. Six ROIs were obtained from each mouse. **(M)** Motion analysis of WT-MuSC–, Δ7-MuSC^Empty–, and Δ7-MuSC^miRs–derived myotubes. **(N)** Maximal SMCV. Four or five mice from each group were evaluated. The number of analyzed ROIs is as follows: WT = 30, Δ7-SMA with miRs = 31, Δ7-SMA without miRs = 31. Error bars indicate means ± SD. **(C, D, E, H, K, N)** Statistical analysis by one-way ANOVA with multiple comparisons (C, D, E) and a t test (H, K, N). Each dot represents a biologically independent sample.

appeared to couple with SMN binding to their promoter regions. Therefore, SMN-mediated transcriptional regulation could be one of the mechanisms for the tissue-specific phenotypes in SMA.

Our results showed that RNAPLII, SMN, and MYOD1 colocalized in the promoter region of several genes, and highlighted the physiological interaction between SMN and MYOD1, suggesting that SMN plays a functional role in MYOD1-driven gene expression. In contrast, ChIP-seq analysis revealed that the interaction of SMN with MYOD1 in the genomic loci was not necessary for all MYOD1-bound genes to regulate the expression. This indicates that the binding of SMN on the genomic loci is not dependent on MYOD1. To elucidate the detailed mechanism of SMN-mediated myogenic gene expression, we need to examine the factors and modifications that determine the binding of SMN to promoter regions, and a functional assessment of SMN on MYOD1. We also showed that the binding of SMN at the promoter region of MYOD1, miR-1, and miR-206 regulates the MYOD1, miR-1, and miR-206 transcriptional levels. SMA motor neurons have been reported to show alternations in miR-1 and miR-206 expression (Wang et al, 2014; Luchetti et al, 2015; Wertz et al, 2016). In addition, SMN protein binds to miR processing proteins including fragile X mental retardation protein, KH-type splicing regulatory protein, and fused in sarcoma/translocated in liposarcoma (Piazzon et al, 2008; Tadesse et al, 2008; Yamazaki et al, 2012). Therefore, it is thought that SMN deficiency disturbs miR processing, thus altering miR expression. However, there is no evidence that SMN protein has a role in the miRNA processing complex. Though we did not show whether SMN protein contributes to miR processing in myogenic differentiation, our results proved that SMN protein regulates the expression of miRs by regulating the promoter activity of the host genes. In our study, we found that transcriptional regulation is mediated by SMN during myogenic differentiation. Until now, there has been no report showing the SMN-mediated transcriptional regulation of motor neurons. The effects of SMN on transcription during motor neurogenesis warrant further study.

We found that the nuclear localization of SMN was transient, indicating that it binds to specific genes only. The introduction of miR-1 and miR-206 into MuSCs derived from Δ7-SMA mice improved muscle function and myotube formation. This result indicates that the postnatal introduction of miR-1 and miR-206 benefits MuSCs in SMA treatment. Supporting this finding, the miR-1– and miR-206–mediated pathway is impaired in SMN-depleted satellite cells (Qing Liu, 1997).

SMA model mice have fewer Pax7[+] MyoD1[-] satellite cells and a lower capacity to regenerate damaged muscle (Hayhurst, 2012; Kim, 2020), but how SMN deficiency triggers the pathology in satellite cells is unknown. Our results revealed that SMN could regulate the transcription of initial myogenic differentiation factors including MyoD1, miR-1, and miR-206. These results suggested that the disturbed initial myogenic transcriptional network by SMN deficiency could impair the maintenance of satellite cells and their capacity to differentiate. To elucidate when SMN is required and what function it has in satellite cells, we should analyze the sequential transcriptional changes in satellite cells derived from Δ7-SMA mice after differentiation and validate whether SMN nuclear localization in satellite cells corresponds to the myogenic differentiation signal.

Recently, several new drugs targeting SMN have been marketed and shown significant improvement in the prognosis of type I SMA (Passini et al, 2010; Mendell et al, 2017). However, even with early intervention with these drugs, significant functional impairment remains in most of the cases, making it difficult to catch up with normal development (Finkel et al, 2016, 2017; Mendell et al, 2017; De Vivo et al, 2019). Therefore, although therapies that increase full-length SMN expression have a significant impact on the disease course of SMA and the quality of life of patients, further functional improvement is needed to reduce the burden of the disease. One way to improve motor function in SMA patients is to develop therapies that prevent the loss of skeletal muscle in SMA patients and combine them with the restoration of SMN in motor neurons (Long et al, 2019). The regulatory function of SMN for miRs that we have discovered may contribute to the development of such a therapy. Therefore, we plan to investigate whether miR administration improves prognosis in Δ7-SMA mice.

In conclusion, the down-regulation of miR-1 and miR-206 caused mitochondrial dysfunction in the skeletal muscle of SMA models. Our results indicated that SMN is the modulator for transcription during a specific phase of differentiation by binding to genome loci. Our results further suggest miR-1 and miR-206 are candidate therapeutic targets in SMA.

# Materials and Methods

### Resource availability

#### Material availability
iPSCs and other research reagents generated by the authors will be distributed upon request to other researchers.

### Dataset availability

ChIP-seq data have been deposited in NCBI SRA under accession number PRJNA895192.

### Resources and reagents

Details of the resources and reagents used in this study are described in Table 1.

### Experimental model and subject details

#### Ethics statement
The iPSC study was approved by the Ethics Committees of Kyoto University (R0091/G0259). Written informed consent was obtained from the patients or their guardians in accordance with the Declaration of Helsinki. The study plan for recombinant DNA research was approved by the Recombinant DNA Experiment Safety Committee of Kyoto University. Animal studies were approved by the institutional review board. All methods were performed in accordance with the relevant guidelines and regulations.

**Table 1.  Resources and reagents used in this study.**

| Reagent or Resource | Source | Identifier |
|---|---|---|
| Antibodies | | |
| Mouse monoclonal anti-SMN | BD Transduction Laboratories | Cat# 610647; RRID: AB_397973 |
| Rabbit polyclonal anti-MT-ND1 | Abcam | Cat# ab222892; RRID: AB_ |
| Rabbit monoclonal anti-MTCO1 (EPR19628) | Abcam | Cat# ab203912; RRID: AB_2801537 |
| Mouse monoclonal anti-Fast Myosin Skeletal Heavy chain (MY-32) | Abcam | Cat# ab51263; RRID: AB_2297993 |
| Mouse monoclonal anti-MyoD1 (5.2F) | Abcam | Cat# ab16148; RRID: AB_2148758 |
| Mouse monoclonal anti-MTCO2 (12C4F12) | Abcam | Cat# ab110258; RRID: AB_10887758 |
| Mouse monoclonal anti-Sarcomeric Alpha Actinin (EA-53) | Abcam | Cat# ab9465; RRID: AB_307264 |
| Rat monoclonal anti-RNA polymerase II CTD repeat YSPTSPS (1C7) | Abcam | Cat# ab252854 |
| Rabbit monoclonal MyoD1 (D8G3) XP | Cell Signaling Technology | Cat 13812S; RRID: AB_2798320 |
| Rabbit monoclonal cleaved caspase-3 (Asp175) (5A1E) | Cell Signaling Technology | Cat# 9664S; RRID: AB_2070042 |
| Rabbit polyclonal caspase-3 | Cell Signaling Technology | Cat# 9662S; RRID: AB_331439 |
| Rabbit polyclonal VDAC | Cell Signaling Technology | Cat# 4866S; RRID: AB_2272627 |
| Rabbit monoclonal GAPDH (14C10) | Cell Signaling Technology | Cat# 2118S; RRID: AB_561053 |
| Rabbit monoclonal Histone H3 (D2B12) XP (ChIP Formulated) | Cell Signaling Technology | Cat# 4620S; RRID: AB_1904005 |
| Mouse monoclonal His-Tag (27 × 10$^8$) | Cell Signaling Technology | Cat# 2366S; RRID: AB_2115719 |
| Rabbit monoclonal DYKDDDDK Tag (D6W5B) | Cell Signaling Technology | Cat# 14793S; RRID: AB_2572291 |
| Rabbit monoclonal $\beta$-actin (13 × 10$^5$) | Cell Signaling Technology | Cat# 5125S; RRID: AB_1903890 |
| Rabbit polyclonal anti-rabbit IgG, HRP-linked antibody | Cell Signaling Technology | Cat# 7074S; RRID: AB_2099233 |
| Horse polyclonal anti-mouse IgG, HRP-linked antibody | Cell Signaling Technology | Cat# 7076S; RRID: AB_330924 |
| Rabbit monoclonal (DA1E) IgG XP isotype Control | Cell Signaling Technology | Cat# 3900S; RRID: AB_1550038 |
| Mouse monoclonal anti-rabbit IgG (conformation specific) (L27A9) (HRP conjugate) | Cell Signaling Technology | Cat# 5127S; RRID: AB_10892860 |
| Rabbit monoclonal cleaved caspase-3 (Asp175) (D3E9) rabbit (Alexa Fluor 647 conjugate) | Cell Signaling Technology | Cat# 9602S; RRID: AB_2687881 |
| Anti-mouse IgG (H + L), F(ab')2 fragment (Alexa Fluor 647 conjugate) | Cell Signaling Technology | Cat# 4410S; RRID: AB_1904023 |
| Anti-mouse IgG (H + L), F(ab')2 fragment (Alexa Fluor 488 conjugate) | Cell Signaling Technology | Cat# 4408S; RRID: AB_1904023 |
| Anti-mouse IgG (H + L), F(ab')2 fragment (Alexa Fluor 594 conjugate) | Cell Signaling Technology | Cat# 4412S; RRID: AB_1904025 |
| Anti-rabbit IgG (H + L), F(ab')2 fragment (Alexa Fluor 647 conjugate) | Cell Signaling Technology | Cat# 8890S; RRID: AB_2714182 |
| Anti-rabbit IgG (H + L), F(ab')2 fragment (Alexa Fluor 488 conjugate) | Cell Signaling Technology | Cat# 4414S; RRID: AB_10693544 |
| Rabbit IgG isotype control (Alexa Fluor 647 conjugate) | Cell Signaling Technology | Cat# 3452S; RRID: AB_10695811 |
| Mouse (MOPC-21) mAb IgG$_1$ isotype control (Alexa Fluor 647 conjugate) | Cell Signaling Technology | Cat# 4843S; RRID: AB_1281292 |
| Negative control mouse IgG$_1$ | Dako | Cat# X0931 |
| $\alpha$-Bungarotoxin, Alexa Fluor 647 Conjugate | Molecular Probes | Cat# B35450 |
| Neuronal class III $\beta$-tubulin (TUJ1) | Covance | Cat# MMS435P; RRID: AB_2313773 |
| Anti-synaptic vesicle glycoprotein 2A | DSHB | Cat# SV2; RRID: AB_2315387 |
| Myogenin antibody (MGN185) Alexa Fluor 647 | NOVUS | Cat# NBP2-33056AF647 |

| Reagent or Resource | Source | Identifier |
|---|---|---|
| Integrin α7 antibody, anti-mouse, APC | Miltenyi Biotec | Cat# 130-123-833; RRID: AB_2889699 |
| PE rat anti-mouse Ly-6A/E (Sca-1) | BD Biosciences | Cat# 553336; RRID: AB_394792 |
| Rat anti-CD31 monoclonal antibody, phycoerythrin-conjugated, clone MEC 13.3 | BD Biosciences | Cat# 553373; RRID: AB_394819 |
| PE anti-mouse CD45 antibody | BioLegend | Cat# 147712; RRID: AB_2563598 |
| Bacterial and virus strains | | |
| pcDNA3.1 + SMN1 myc HIS | Unpublished | Addgene; Cat# 71687 |
| FLAG-hMYOD1 | Unpublished | Addgene; Cat# 78329 |
| Mouse pre-microRNA expression construct mir1a-1 | System Biosciences | Cat# MMIR-1a-1-PA-1 |
| Mouse pre-microRNA expression construct mir206 | System Biosciences | Cat# MMIR-206-PA-1 |
| Chemicals, peptides, and recombinant proteins | | |
| 3,3′- Diethyloxacarbocyanine iodide | Life Technologies | Cat# M3415 |
| CellROX green flow cytometry assay kit | Molecular Probes | Cat# C10492 |
| MitoSOX red mitochondrial superoxide indicator | Invitrogen | Cat# M36008 |
| Caspase inhibitor Z-VAD-FMK, 20 mM | Promega | Cat# G7231 |
| α-Tocopherol | Sigma-Aldrich | Cat# T3251-5G |
| Dynabeads Sheep Anti-Rat IgG | Invitrogen | Cat# 11035 |
| Dynabeads M-280 Sheep Anti-Rabbit IgG | Invitrogen | Cat# 11203D |
| Dynabeads M-280 Sheep Anti-Mouse IgG | Invitrogen | Cat# 11201D |
| Dynabeads His-Tag Isolation & Pulldown | Invitrogen | Cat# 10103D |
| Experimental models: organisms/strains | | |
| FVB.Cg-Tg(SMN2) 89Ahmb Smn1$^{tm1Msd}$Tg (SMN2*Δ7) 4299AhmB/J | Jackson Laboratories | Cat# 007951; RRID: IMSR_JAX:007951 |
| Oligonucleotides | | |
| See also Table S2 | | |
| Primer 1 for the genotyping: Forward: CGGCTACCTGCCCATTCGACCACC | Le et al (2005) | N/A |
| Primer 1 for the genotyping: Reverse: CCTTAAAGGAAGCCACAGCTTTATC | Le et al (2005) | N/A |
| Primer 2 for the genotyping: Forward: TCCAGCTCCGGGATATTGGGATTG | Le et al (2005) | N/A |
| Primer 2 for the genotyping: Reverse: AGGTCCCACCACCTAAGAAAGCC | Le et al (2005) | N/A |
| Primer for U6 forward | TaKaRa Bio | Cat# 638315 |
| Primer for U6 reverse | TaKaRa Bio | Cat# 638315 |
| Primer for mRQ 3′ primer | TaKaRa Bio | Cat# 638315 |
| Software and algorithms | | |
| GraphPad Prism 6 | GraphPad Software, Inc. | N/A |
| Cutadapt (version 1.15) | Martin (2011) | RRIS:SCR_011841 |
| bowtie2 (version 2.2.5) | Langmead and Salzberg (2012) | RRIS:SCR_016368 |
| MACS2 (version 2.2.1.20160309) | Zhang et al (2008) | RRIS:SCR_013291 |
| ImageJ | National Institutes of Health | https://imagej.nih.gov/ij/; RRIS:SCR_003070 |

## Human iPSC lines

A human iPSC line, 201B7, was kindly provided by Dr. Shinya Yamanaka (Kyoto University). Isogenic iPSC lines with doxycycline-inducible MYOD1 construct (B7-M and B7-MSMNKD) were established in a previous study (Lin et al, 2019). The iPSC line established from a type I SMA patient was also established in a previous study (Yoshida et al, 2015). The doxycycline-inducible MYOD1 overexpression vector (Lin et al, 2019) was introduced into SMA patient iPSCs by FuGENE HD (Promega), and iPSCs that stably expressed the vector (SMA-M) were selected with G418 (Wako) (50 µg/ml). A constitutive SMN1 expression vector was introduced into SMA-M to establish the SMA-M$^{OE}$ clone, and SMA-M$^{OE}$ was selected with puromycin (InvivoGen) (1.0 µg/ml). For ChIP-seq analysis of MYOD1, we established an iPSC line with a doxycycline-inducible FLAG-MYOD1 construct. The sequence of FLAG-MYOD1 (#78329; Addgene) was cloned into the doxycycline-inducible overexpression vector. Then, the doxycycline-inducible FLAG-MYOD1 construct was introduced into 201B7. Lastly, a stable clone was selected using puromycin (1.0 µg/ml).

## Conversion of hiPSC clones into myogenic cells

The iPSC lines with doxycycline-inducible MYOD1 expression vector (B7-M, B7-MSMNKD, SMA-M, and SMA-MOE) were converted into myogenic cells as previously described (Tanaka, 2013). In brief, 4.0 × 10$^5$ iPSCs were seeded onto Matrigel (Corning)-coated 24-well plates in Primate ES Cell Medium (ReproCELL). At day 0, the medium was exchanged with primate ES Cell Medium containing doxycycline (TaKaRa) (1.0 µg/ml). From day 1, the medium was exchanged everyday with Minimum Essential Medium A (Gibco) containing 10% KnockOut Serum Replacement (Gibco) and doxycycline (1.0 µg/ml). Cells were collected with Accumax (Nacalai Tesque) on days 0, 3, and 6 after the myogenic conversion and analyzed thereafter.

## C2C12 culture and differentiation

A mouse myoblast cell line, C2C12, was kindly provided by Dr. Atsuko Sehara-Fujisawa (Kyoto University) and maintained with Dulbecco's Minimal Essential Medium (DMEM) (Nacalai Tesque) containing 10% FCS (Sigma-Aldrich). To differentiate the cells into myotubes, the culture medium of confluent C2C12 cells was replaced with DMEM containing 2% horse serum (HS; Sigma-Aldrich). For subsequent analysis, the cells were collected with 0.05% trypsin–EDTA (Gibco) at days 3 and 6.

## siRNA and microRNA mimic transfection

siSmn (SASI_Mm01_00155410; Sigma-Aldrich), miRNAs mimicking miR-1 and miR-206 (Gene Design), or scramble negative control (Gene Tools, LLC, Standard control) (final concentration, 100 nM) was transfected into C2C12 cells seeded onto 24- or 96-well plates using Lipofectamine RNAiMAX (Life Technologies) after the cells reached 70% confluency according to the manufacturer's instruction.

## RNA isolation and quantitative PCR (qRT-PCR)

For the RT–qRT-PCR analysis of mRNA, total RNA was extracted from the cells with the RNeasy Mini Kit (QIAGEN), and RT was performed using the PrimeScript RT Master Mix (TaKaRa). For the RT–qRT-PCR analysis of miR, total RNA was extracted from cells with the miR-Neasy Mini Kit (QIAGEN), and complementary DNA of miR was synthesized using the Mir-X miRNA First-Strand Synthesis Kit (TaKaRa). RT–qRT-PCR was performed with TB Green Premix Ex Taq II (Tli RNaseH Plus) (TaKaRa) on the StepOnePlus Real-Time PCR System (Applied Biosystems) according to the manufacturer's protocol. Ribosomal protein L13a or U6 was used as the internal control. Primer sequences are listed in Table S2.

## Intracellular flow cytometry analysis

iPSC-derived myogenic cells on day 3 were collected and incubated with 0.2% saponin (Nacalai Tesque) and 4% paraformaldehyde (Nacalai Tesque) on ice for 5 min for permeabilization and fixation. Then, the cells were labeled with antibodies against cleaved caspase-3 (1:20; #9602S; CST) or MyoG (1:20; #NBP2-33056; Novus Biologicals). The following isotype controls were used: rabbit IgG Isotype Control (Alexa Fluor 647 Conjugate) (1:20; #3452S; CST) and mouse (MOPC-21) mAb IgG$_1$ Isotype Control (Alexa Fluor 647 Conjugate) (1:20; #4843S; CST). Antibodies were incubated with cells for 90 min at room temperature. The labeled cells were analyzed with BD FACSAria (BD Biosciences), and the results were analyzed and processed with FlowJo software (Tree Star Inc.).

## Measurement of reactive oxygen species and membrane potential (△ψm)

For the measurement of the total reactive oxygen species level, the cells were incubated with a medium containing CellROX Green reagent (Invitrogen) (2.5 µM) for 60 min at 37°C. Cells were then washed three times with PBS. For mitochondrial superoxide production analysis, cells were incubated with a medium containing MitoSOX Red mitochondrial superoxide indicator reagent (Invitrogen) (2.5 µM) for 60 min at 37°C and then washed three times with PBS. To measure mitochondrial membrane potential (Δψm), the cells were incubated with a medium containing the 3,3'-diethyloxacarbocyanine iodide (5.0 µM) (Life Technologies) for 45 min at 37°C and then washed three times with PBS. The stained cells were analyzed with BD FACSAria, and the results were analyzed and processed with FlowJo software.

## Immunocytochemistry

Cells on a multi-well glass-bottom dish (D141400; MATSUNAMI) were washed three times with PBS and incubated with ice-cold methanol for 15 min at −30°C. Fixed cells were further washed three times with PBS and incubated with 5% BSA (Sigma-Aldrich) in PBS for 30 min at room temperature. Primary antibody reactions were performed at 4°C overnight. Secondary antibody reactions were performed at room temperature for 90 min. Stained cells were washed three times with PBS containing DAPI (Sigma-Aldrich). The image was taken with a FLUOVIEW FV1000 (Olympus). The following primary

and secondary antibodies were used: anti-SMN (1:1,000; #610647; BD Transduction Laboratories), anti-Fast Myosin Skeletal Heavy chain (1:1,000; #ab51263; Abcam), anti-Sarcomeric A Actinin (1:1,000; #ab9465; Abcam), anti-mouse IgG (H + L), F(ab′)2 Fragment (Alexa Fluor 488 Conjugate) (1:1,000; #4408S; CST), anti-rabbit IgG (H + L), F(ab′)2 Fragment (Alexa Fluor 488 Conjugate) (1:1,000; #4412S; CST), anti–synaptic vesicle protein 2 (1:50; SV2; DSHB), anti-Tuj1 (1:1,000; MAB1195; R&D Systems), and Alexa Fluor 647–conjugated α-bungarotoxin (0.5 µg/ml, B3545; Molecular Probes).

### Image analysis

To show myotube formation, the ratio of nuclei labeled in the DAPI-positive area to myosin heavy chain–positive area was processed and analyzed with ImageJ (NIH). To show the signal intensity of SMN in the nuclei, the nuclear area was recognized by the DAPI-positive area, and the signal intensity of SMN in the nuclei excluding SMN foci was quantified with ImageJ.

### Immunoblotting

Cells were collected with Accumax and centrifuged at 4,400$g$ for 15 s at 4°C. To extract proteins, cell pellets were lysed with RIPA buffer (Wako) and incubated for 30 min on ice. The lysate was centrifuged at 21,900$g$ for 15 min at 4°C. The supernatant was mixed with 2 × Laemmli sample buffer (Bio-Rad Laboratories) containing 5% total volume of 2-mercaptoethanol (Nacalai Tesque) and boiled for 5 min at 95°C. Polyacrylamide gel electrophoresis was performed on SDS–polyacrylamide gels, and proteins were transferred to a nitrocellulose membrane (Merck Millipore). The membrane was then incubated with 5% BSA in Tris-buffered saline with Tween-20 (Santa Cruz Biotechnology, Inc.) for blocking. The primary antibody reaction was performed at 4°C overnight. The secondary antibody incubation was performed for 90 min at room temperature, and then, the protein was detected using ECL chemiluminescence reagents (Thermo Fisher Scientific). Antibody against $\beta$-actin was reacted for 60 min at room temperature. The following primary and secondary antibodies were used: anti-SMN (1:1,000; #610647; BD Transduction Laboratories), anti-ND1 (1:1,000; #ab222892; Abcam), anti-COX1 (1:1,000; #ab51263; Abcam), anti-MYOD1 (1:1,000; #13812S; CST), anti-cleaved caspase-3 (1:1,000; #9664S; CST), anti-caspase-3 (1:1,000; #9662S; CST), anti-VDAC (1:1,000; #4866S; CST), anti-His-Tag (1:1,000; #2366P; CST), anti-GAPDH (1:1,000; #2118S; CST), anti-Histone H3 (1:1,000; #4620S; CST), anti-Flag (1:1,000; #14793S; CST), anti-COX2 (1:1,000; #ab110258; Abcam), anti-Sarcomeric A Actinin (1:1,000; #ab9465; Abcam), anti-$\beta$-actin (1:5,000; #5125S; CST), anti-mouse-HRP (1:2,500; #7076S; CST), anti-rabbit HRP (1:2,500; #7074S; CST), anti-rabbit (Conformation Specific) HRP (1:2,500; #5127S; CST), and anti-mouse (Conformation Specific) HRP (1:2,500; #ab131368; Abcam).

### Plasmid construction and reporter activity assay

Mouse $\beta$-actin and the distal and proximal linc-MD1 promoter sequences were amplified from genomic DNA extracted from the mouse C2C12 cell line using PCR. Then, each promoter sequence was cloned into upstream of GFP coding region. For the reporter assay, C2C12 cells were seeded onto a 24-well plate in DMEM supplemented with 10% FCS. On day 0, the reporter plasmid with each promoter sequence was transfected into C2C12 cells using Lipofectamine 3000 (Life Technologies) after the cells reached 70% confluency, according to the manufacturer's instructions. On day 1, siSmn or scramble negative control (final concentration, 100 nM) was transfected into C2C12 cells, as previously described. On day 2, the media were exchanged with DMEM supplemented with 10% FCS. Then, on day 3, to induce differentiation into myotubes, the media were exchanged into DMEM supplemented with 2% HS. Finally, on the next day, the GFP signal intensity was measured using flow cytometry.

### Mice

Δ7 SMA mice (Le et al, 2005) were purchased from Jackson Laboratories (FVB.Cg-Tg [SMN2] 89Ahmb Smn1$^{tm1Msd}$Tg [SMN2*Δ7] 4299AhmB/J; stock no. 005025). Primers for genotyping were designed as described previously (Le et al, 2005). All experiments were performed on P3 or P5 mice. Smn$^{+/+}$; SMN2$^{+/+}$; SMNΔ7$^{+/+}$ was used as WT. Mice were maintained at the animal facility according to an institutionally approved protocol.

### Isolation and primary culture of MuSCs

Isolation and culture of MuSCs were performed as previously described (Hayhurst et al, 2012). Briefly, skeletal muscle tissue was isolated from the TA and GA of neonatal mice. The muscle tissue was shredded by sterilized scissors, incubated with 0.2% (wt/vol) collagenase II (Roche) in DMEM containing 20% FCS and antibiotic–antimycotic (100×) (Gibco) for 30 min at 37°C, and dissociated into single myofibers by pipetting several times. The dissociated tissue was then centrifuged at 4,400$g$ for 15 s and suspended with 20% FCS DMEM. The cell suspension including MuSCs was seeded onto a six-well plate coated with Matrigel. The attached cells including MuSCs were then expanded in 20% FCS DMEM condition for a few days. After reaching confluence, the MuSCs were cultured to induce myotubes in DMEM containing 2% HS for 3 d.

### TEM

TEM sample preparation and analysis were performed following a previous protocol (Lin et al, 2019). The TA, GA, and diaphragm were dissected from Δ7-SMA or WT mice. The tissues were cut into small pieces of about 2 mm$^2$ and then fixed by 0.1 M phosphate buffer (pH 7.4) including 2% PFA (Electron Microscopy Sciences) and 2% glutaraldehyde (Electron Microscopy Sciences) overnight at 4°C. Postfixation was carried out in 1% osmium tetroxide solution (Electron Microscopy Sciences) for 1 h at room temperature. The samples were dehydrated in graded concentrations of ethanol (30%, 50%, 70%, 90%, 95%, and 100%) and embedded in Epon resin (Electron Microscopy Sciences). Ultrathin sections (80 nm) were cut and stained with uranyl acetate and alkaline lead citrate. The specimens were examined with a TEM (H-7650; Hitachi).

## Motion vector analysis

Motion quantification in an ROI was performed using the SI8000 Cell Motion Imaging System (Sony) as previously described (Lin et al, 2019). Both moving image capture and motion analyses were performed using this system. To compare muscle contraction ability between WT and Δ7-SMA MuSC-derived myotubes, moving images were acquired under continuous square-wave electric current stimulation (25 V) using NEPA21 (Nepa Gene). To show muscle contraction ability, the maximal SMCV, which indicates the maximal value of the SMCV during electric current stimulation, was applied.

## Production and infection of lentiviral vectors

To produce the virus particle, a lentivirus expression plasmid (MMIR-1a-1-PA-1 and MMIR-206-PA-1 purchased from System Biosciences) and plasmid mixture for packaging (ViraPower HiPerform Lentiviral Expression Systems; Invitrogen) were transfected into 293 package cells with Lipofectamine 2000 (Invitrogen) under DMEM (10% FCS) condition. After incubation for 2 d, the medium containing the virus particles was recovered and concentrated with Polyethylene Glycol 8000 (Sigma-Aldrich) at 4°C overnight. The medium was centrifuged at 2,000$g$ for 30 min. The virus particle pellets were suspended with PBS. For virus infection into C2C12 cells or MuSCs, virus solution was added into the cell suspension in a 1.5-ml tube, and then, the mixture was incubated for 60 min at 37°C. After the incubation, the mixture was seeded onto culture ware.

## Immunoprecipitation (IP)

IP was performed with Dynabeads (Thermo Fisher Scientific) according to the manufacturer's protocol. The preincubation of Dynabeads (100 μl) (M-280 Sheep Anti-Mouse IgG or M-280 Sheep Anti-Rabbit IgG) with primary antibody (5.0 μl) in 1.0% BSA/PBS was performed at 4°C overnight. The beads conjugated with primary antibodies were then washed with IP buffer (10 mM Tris–HCl [pH 7.8] [Nacalai Tesque], 1.0% NP-40 [Nacalai Tesque], and 15 mM NaCl [Nacalai Tesque]/EDTA-free protease inhibitors [100×] [Nacalai Tesque]). Cells were collected with a cell scraper in IP buffer. A total of 1.0 × 10$^7$ cells were sonicated on ice three times for 1 s by the XL-2000 sonicator (MISONIX). Sonicated cells were centrifuged at 21,900$g$ for 10 min. The supernatant was collected and incubated with primary antibody conjugated with Dynabeads at 4°C overnight. After incubation, the sample tubes were set on DynaMag-2 (Thermo Fisher Scientific), and Dynabeads were washed three times with IP buffer. Dynabeads were then suspended with fresh IP buffer and incubated at 95°C for 5 min. The supernatant was collected on DynaMag-2 and mixed with 2 × Laemmli sample buffer. Immunoblotting was performed as described.

## Chromatin immunoprecipitation (ChIP)–quantitative PCR (qRT-PCR) sequencing (seq)

The preincubation of Dynabeads (100 μl) (M-280 Sheep Anti-Mouse IgG; Thermo Fisher Scientific) with primary antibody (5.0 μg) was performed at 4°C overnight. For ChIP-seq, preincubation was performed in the presence of 40 μg of salmon sperm DNA (Thermo

Fisher Scientific). For ChIP–qRT-PCR and ChIP-seq, 1.0 × 10$^7$ cells were collected and then cross-linked in 1.0% (wt/vol) formaldehyde solution for 30 min at room temperature. Cross-linked cells were neutralized with glycine (Wako) and then centrifuged at 1,200$g$ for 2 min. Cell pellets were suspended with 2.0% FSC/PBS, rotated for 10 min at 4°C, and then centrifuged at 1,200$g$ for 2 min. In the lysis procedure, cell pellets were suspended with lysis buffer 1 (50 mM Hepes buffer [Hampton Research], 140 mM NaCl [Nacalai Tesque], 1.0 mM EDTA [Nacalai Tesque], 10% glycerol [Wako], 0.5% NP-40, and 0.25% Triton X-100 [Thermo Fisher Scientific]) and rotated for 10 min at 4°C. After centrifugation at 16,400$g$ for 5 min, the cell pellets were suspended in lysis buffer 2 (10 mM Tris–HCl [Nacalai Tesque], 200 mM NaCl, 1.0 mM EDTA, and 0.5 mM EGTA [Nacalai Tesque]) and rotated for 10 min at 4°C. The cell suspension was then centrifuged at 16,400$g$ for 5 min, and lysis buffer 2 was then replaced with lysis buffer 3 (10 mM Tris–HCl, 100 mM NaCl, 1.0 mM EDTA, 0.5 mM EGTA, 0.1% sodium deoxycholate [Wako], and 0.5% N-lauroylsarcosine [Nacalai Tesque]). The fragmentation of cross-linked DNA in lysis buffer 3 was performed by SFX250 sonicator (BRANSON). Fragmented DNA was incubated with Dynabeads at 4°C overnight. After the incubation, the samples were washed four times with RIPA buffer (50 mM Hepes buffer, 500 mM LiCl [Sigma-Aldrich], 1.0 mM EDTA, 1.0% NP-40, and 0.7% sodium deoxycholate) on DynaMag-2 and then eluted. To detach the antibodies from Dynabeads, Dynabeads were incubated at 65°C for 15 min, and the supernatant was collected on DynaMag-2. Cross-linking was reversed for 24 h at 65°C. Reverse cross-linked samples were purified using ChIP DNA Clean & Concentrator (ZYMO Research) according to the manufacturer's protocol. Eluted DNA was used as the template for qRT-PCR. The primer sequences used for ChIP–qRT-PCR are listed in Table S2. To generate DNA libraries for ChIP-seq, eluted DNA was fragmented using a Covaris Focused-ultrasonicator M220 (M&S Instruments Inc.). The library was prepared using a SMARTer ThruPLEX DNA-seq 48S kit (TaKaRa) and sequenced on a NextSeq 500 System (Illumina) using 75-bp single-end reads. The ChIP-seq reads were trimmed using Cutadapt (version 1.15) and mapped in bowtie2 (version 2.2.5) to hg19 after removing the reads mapped to salmon. For visualization, ChIP peaks were called and normalized by the number of mapped reads using MACS2 (version 2.2.1.20160309) using the input reads as the control. To identify ChIP peaks, the findPeaks program in HOMER was used (Heinz et al, 2010). For RNAPLII and SMN ChIP-seq, the program was used with the -region option, the minimum distance of peaks was set to 100 bp, and the local fold change cutoff was disabled. Similarly, for FLAG-MyoD1 ChIP-seq, both the peak size and minimum distance of peaks were set to 500 bp, the local fold change cutoff was disabled, and the default settings were applied for all other parameters. The genomic annotation of ChIP-seq peaks was conducted using the annotatePeaks program in HOMER. Heatmaps and signal density histograms were generated using ngs.plot (version 2.61) (Shen et al, 2014).

## Mitochondrial oxygen consumption rate

The mitochondrial OCR was measured using an XF96 Extracellular Flux Analyzer (Seahorse Bioscience) according to the manufacturer's instruction. C2C12 cells (1.0 × 10$^4$ cells) were seeded onto XF96 Cell Culture Microplates (Agilent Technologies) coated with

Matrigel. C2C12 cells were expanded in 10% FCS DMEM for 2 d and then differentiated into myotubes in 2% HS DMEM. The differentiated media were exchanged at days 3, 5, and 6. On day 5, a total of $1.0 \times 10^4$ iPSC-derived myogenic cells were replated onto XF96 Cell Culture Microplates coated with Matrigel, and then analyzed on day 6. Next, WT or Δ7-SMA mouse-derived CD45⁻ Sca-1⁻ CD31⁻ Integrin α7⁺ cells were infected with lentivirus containing miRs or an empty vector. The cells ($1.0–1.5 \times 10^4$) were then seeded onto XF96 Cell Culture Microplates coated with Matrigel. Primary myogenic cells were expanded in 10% FCS/low-glucose DMEM supplemented with FGF-2 (2.5 ng/μl) (#NBP2-76182; Novus Biologicals) before moving to 2% HS DMEM for the induction of cell differentiation into myotubes. Mitochondrial OCR was measured using primary myogenic cells and C2C12 cells. To this end, the analyzed cells were detached and counted using a hemacytometer after analysis. The value of oxygen consumption in the primary myogenic cells and C2C12 cells was normalized according to the cell number (pmol/min/10,000 cells). Before the analysis, the culture medium was replaced with DMEM (Sigma-Aldrich) containing 5.0 mM glucose, 5.0 mM GlutaMAX (Gibco), and 1.0 mM sodium pyruvate (Gibco). The cells were incubated for 60 min at 37°C without $CO_2$. To measure the mitochondrial function, the following compounds were added: 8–10 μM oligomycin (Sigma-Aldrich), 10 μM carbonyl cyanide-p-(trifluoromethoxy)phenylhydrazone (FCCP; Sigma-Aldrich), and 1.0 μM antimycin A (Sigma-Aldrich) and 1.0 μM rotenone (Sigma-Aldrich). Each mitochondrial oxygen respiration–related value was calculated using Wave software (Agilent Technologies).

## Supplementary Information

## Acknowledgements

We thank Ms. Harumi Watanabe for providing administrative assistance, Dr. Peter Karagiannis for proofreading the article, Dr. Misaki Ouchida for graphical assistance, and Drs. Shiori Ando and Hideaki Hara for technical support. This work was supported by grants from the Japan Society for the Promotion of Science KAKENHI Grant Numbers 16H0552 (MK Saito) and 20H03642 (MK Saito), the Core Center for iPS Cell Research of Research Center Network for Realization of Regenerative Medicine from Japan Agency for Medical Research and Development JP21bm0104001 (T Nakahata and MK Saito), and the iPS Cell Research Fund (MK Saito).

### Author Contributions

A Ikenaka: conceptualization, data curation, investigation, visualization, and writing—original draft.
Y Kitagawa: data curation, investigation, visualization, methodology, and writing—original draft.
M Yoshida: conceptualization, investigation, and writing—review and editing.
C-Y Lin: investigation, visualization, and writing—review and editing.
A Niwa: data curation, supervision, and writing—review and editing.

T Nakahata: data curation, supervision, and writing—review and editing.
MK Saito: conceptualization, supervision, funding acquisition, project administration, and writing—review and editing.

### Conflict of Interest Statement

The authors declare that they have no conflict of interest.

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
