## [Reviewer comments · Life Science Alliance]

Life Science Alliance

SMN promotes mitochondrial metabolic maturation during myogenesis by regulating the MYOD-miRNA axis.

Akhiro Ikenaka, Yohko Kitagawa, Michiko Yoshida, Chuang-Yu Lin, Akira Niwa, Tatsutoshi Nakahata, and Megumu Saito
DOI: <https://doi.org/10.26508/lsa.202201457>

Corresponding author(s): Megumu Saito, Kyoto University

Review Timeline:

Submission Date:	2022-03-19
Editorial Decision:	2022-03-21
Revision Received:	2022-10-30
Editorial Decision:	2022-11-25
Revision Received:	2022-12-12
Accepted:	2022-12-13

Transaction Report:

Please note that the manuscript was reviewed at Review Commons and these reports were taken into account in the decision-making process at Life Science Alliance.

Revision Plan

Manuscript number: RC-2021-01144

Corresponding author(s): Megumu K, Saito

1. General Statements [optional]

We appreciate the reviewers' fruitful comments and suggestions.

2. Description of the planned revisions

Reviewer#1

Comment 4, There is a massive increase in the detection of miRNA host gene RNA between day 3 and day 6, however there is almost no change in the expression of the mature miRNAs. This indicates (beside potential technical shortcomings), that there are massive differences in the processing of the precursor RNAs, thus detection of the host genes or of mature miRNAs, in contrast to the conclusions by the authors, might not reflect regulation of transcriptional activity by a transcription factor. A more direct assay should be used to determine transcriptional activity of the respective genomic loci.

Response : Since the dysregulation of genes associated with miR processing has been reported in SMA motor neurons¹, it is also possible that the post-transcriptional dysregulation of miRs takes place in our case, as the reviewer commented. Accordingly, we amended the description (lines 199-200). However, this report focuses on the behavior of SMN on the genome and the transcription of the miR host gene, not the post-transcriptional regulation. It seems clear from the qPCR data that the transcription of the host gene is indeed affected. However, as the reviewer proposed, it would be better to show the direct effect of SMN on transcription. Therefore, we will perform a reporter assay with the promoter region of the miR host gene.

Comment 5, The maximal oxygen consumption rate depicted in Figure 1 and 3 B/C uses the unit [pmol/min], however the oxygen consumption should be normalized to a defined number of cells or protein content, whatsoever. What is actually the meaning of the additional Scramble=100 in Figure 3B/C? For C2 cells in figure 3B/C the maximal oxygen consumption rate seems to be quite similar to the spare respiratory capacity. Looking at the Oxygen consumption blots, this appears quite unlikely. Depiction of the respective rates, normalized to cell number is needed.

Response : In original Figures 3B and 3C, the value of scramble [pmol/min] was set to 100, and the relative value of each condition to scramble was shown. We apologize for the confusion caused by showing the relative values. In the case of C2C12, unlike iPS cells, the cells tended to fuse strongly and therefore could not be

Revision Plan

reseeded immediately before the OCR assay. Therefore, we adjusted the cell number at the beginning of the differentiation. However, as recommended by the reviewer, we will revise the presentation by normalizing the data with the state of the cells after the assay.

Comment 6, ChIP-PCR experiments were performed to demonstrate binding of SMN to the promoter of MyoD1 and the miR-1 genes to suggest autoregulation of MyoD1 by MYOD1/SMN and regulation of miR-1 genes by MYOD1/SMN. In addition, Co-IPs were performed to demonstrate interaction of SMN1 with MYOD1. To exclude SMN1 might just interact with RNAPL II and thereby affect transcription of multiple genes, CHIP was performed targeting ID3 and Rpl11 promoters and no binding of SMN1 was found for those promoters although SMN interaction with RNAPL II was previously described. This is an interesting and important experiment, however not convincing with the limited number of assays. CHIP-seq of MYOD1, SMN1 and RNAPL II is essential to demonstrate the overlap in DNA binding by MYOD1 and SMN in an unbiased manner, as well as to understand the significance of the SMN - RNAPL II interaction in this model.

Response : We thank the reviewer for recognizing the significance of those experiments. As the reviewer pointed out, demonstrating the overlapping binding site of SMN and MYOD1 is important to understand SMN binding. Therefore, we will perform a ChIP-seq analysis of SMN1 and RNAPLII. For MYOD1, we will use the publicly available dataset for MYOD1 ChIP-seq with C2C12 (GEO accession code: GSM915186) as Reviewer 2, Comment 4, proposed.

Comment 7, To strengthen the findings obtained in C2C12 cells, the authors also analyzed the mitochondrial area in muscle of delta7-SMA mice. Unfortunately, this part of the manuscript is particularly weak. It is not surprising that mice already suffering atrophy of the muscles also reveal changes in mitochondrial content of the muscle. A muscle-specific deletion/modification of SMN would be needed to address the muscle specific function of SMN1.

Response : As the reviewer pointed out, mitochondrial dysregulation in the skeletal muscle of SMA mice is not a novel result. However, we found that, in SMA mice, mitochondrial dysregulation takes place prior to the motor neuron denervation, which was not described in previous reports^{2,3}. Since the *in vivo* data is not an essential part of the overall story, we are thinking of moving it to the supplement. On the other hand, the main objective of this part of the manuscript was to evaluate the function of isolated MuSCs *in vitro*, for which muscle-specific SMN knockout mice are not required. However, we need to perform additional experiments to clearly show the functional difference of SMA mice-derived MuSCs and the effect of miR-1 and -206 on these cells. Therefore, we are planning to evaluate oxygen consumption and the expression level of MyoD1 and miR-1 and -206 in MuSCs isolated from $\Delta 7$ -SMA mice.

Revision Plan

Comment 8, Unfortunately, the analysis of MuSCs isolated from $\Delta 7$ -SMA animals at P5 also is superficial and does not relate to the mechanism suggested in the manuscript. While the authors include data on the contractility of myotubes differentiated from the MuSCs, data on MyoD1 and miRNA expression are missing, no data on the differentiation ability of MuSCs and mitochondrial activity are included. Thus, the miRNA supplementation assay remains incomplete and inconclusive without value to support the model figure 8.

Response : As we described in our response to the previous comment, we will evaluate oxygen consumption and the expression level of MyoD1 and miR-1 and -206 in MuSCs isolated from $\Delta 7$ -SMA mice. For the differentiation ability, we observed improved myotube formation from miRNA-treated $\Delta 7$ -SMA MuSCs.

****Minor comments****

Comment 2, SMN is upregulated in IPS cells overexpressing MyoD1 and in differentiating C2C12 cells. This goes along with increasing amounts of protein in the nucleus. This results do not justify the statement that SMN during myogenesis might have a unique role in the nucleus, because SMN seems to be in the nucleus also prior to differentiation. The figure 4f also should contain total lysate day 0.

Response : We did not mean to conclude that SMN has a unique role in the nucleus during myogenic differentiation from only our observations in figures 4D and S4C.

However, we noticed that SMNs have a characteristic nuclear distribution during differentiation, which is different from typical Cajal body localization, and made a hypothesis accordingly. Therefore, we have amended the text to reduce the strength of the conclusion (lines 235-237). Following the reviewer's suggestion, we are planning to add the total lysate of day 0 in Figure 4F.

Figure A, only for the reviewers. Immunostaining of iPSC-derived myogenic cells on day 3 with SMN.

Revision Plan

Green indicates SMN. The left figure shows typical localization in undifferentiated iPSCs. The right figure shows the transient, diffuse localization observed during myogenesis. White arrows show typical localization of SMN in nuclei.

Reviewer#2

Comment 3, From the experiments in figure 3 it remains unclear whether the decrease in miRNAs directly affects mitochondrial maturation or is a consequence of reduced differentiation of myoblasts to myotubes.

Response: Although we are also interested in this point, it is difficult to clearly distinguish the effect of miRNAs on mitochondrial function and mitochondria-independent myogenesis. In the initial myogenic differentiation, it is known that miR-1 and miR-206 have another target to promote differentiation except for mitochondria, and linc-MD1 also works as competing endogenous RNA in the cytoplasm.⁴⁻⁶ Therefore, the downregulation of SMN could impair both cytoplasmic and mitochondrial pathways during initial myogenic differentiation. Mitochondrial maturation is an important event, but the coordinated differentiation program is essential to promote differentiation^{4,5,7}. In this study, we focused on the MyoD1-miR-mitochondrial maturation pathway, but the contribution of other pathways to SMN muscle maturation will be investigated in future studies. We added these points in the Discussion (lines 351-354).

Comment 4, In Figure 5, SMN ChIP-seq can be performed in C2C12 cells; co-binding of SMN and MyoD can be demonstrated by integrating with existing MyoD ChIP-seq dataset in C2C12 (GEO accession code:GSM915186).

Response : As the reviewer pointed out, demonstrating the overlapping binding site of SMN and MYOD1 is important to understand SMN binding. Therefore, we will perform a ChIP-seq analysis of SMN1 and use the publicly available dataset for MYOD1 ChIP-seq with C2C12 (GEO accession code: GSM915186) as the Reviewer suggested.

Comment 5, In Figure 6, to show the mitochondrial dysfunction in the skeletal muscle of SMA model mice, muscle stem cells from WT and $\Delta 7$ -SMA mice should be isolated to measure the mitochondrial oxygen consumption rate as done in Figure 1E.

Response : According to the reviewer's suggestion, we will isolate muscle stem cells and measure the mitochondrial oxygen consumption rate.

****Minor comments****

Comment 2, In experiments overexpressing miR-1 and miR-206, the overexpression efficiency should be shown.

Response : We will evaluate the expression level of miR-1 and -206. Additionally, we will check the transduction efficiency using flow cytometry.

- 1 Goncalves, I. *et al.* Neuronal activity regulates DROSHA via autophagy in spinal muscular atrophy. *Sci Rep* **8**, 7907, doi:10.1038/s41598-018-26347-y (2018).
- 2 Ripolone, M. *et al.* Impaired Muscle Mitochondrial Biogenesis and Myogenesis in Spinal Muscular Atrophy. *JAMA Neurol* **72**, 666-675, doi:10.1001/jamaneurol.2015.0178 (2015).
- 3 Kim, J. K. *et al.* Muscle-specific SMN reduction reveals motor neuron-independent disease in spinal muscular atrophy models. *J Clin Invest* **130**, 1271-1287, doi:10.1172/JCI131989 (2020).
- 4 Chen, J. F. *et al.* The role of microRNA-1 and microRNA-133 in skeletal muscle proliferation and differentiation. *Nat Genet* **38**, 228-233, doi:10.1038/ng1725 (2006).
- 5 Chen, J. F. *et al.* microRNA-1 and microRNA-206 regulate skeletal muscle satellite cell proliferation and differentiation by repressing Pax7. *J Cell Biol* **190**, 867-879, doi:10.1083/jcb.200911036 (2010).
- 6 Cesana, M. *et al.* A long noncoding RNA controls muscle differentiation by functioning as a competing endogenous RNA. *Cell* **147**, 358-369, doi:10.1016/j.cell.2011.09.028 (2011).
- 7 Shintaku, J. *et al.* MyoD Regulates Skeletal Muscle Oxidative Metabolism Cooperatively with Alternative NF-kappaB. *Cell Rep* **17**, 514-526, doi:10.1016/j.celrep.2016.09.010 (2016).
- 8 Messina, G. *et al.* p27Kip1 acts downstream of N-cadherin-mediated cell adhesion to promote myogenesis beyond cell cycle regulation. *Mol Biol Cell* **16**, 1469-1480, doi:10.1091/mbc.e04-07-0612 (2005).
- 9 Zhang, X. *et al.* MicroRNA directly enhances mitochondrial translation during muscle differentiation. *Cell* **158**, 607-619, doi:10.1016/j.cell.2014.05.047 (2014).
- 10 Wust, S. *et al.* Metabolic Maturation during Muscle Stem Cell Differentiation Is Achieved by miR-1/133a-Mediated Inhibition of the Dlk1-Dio3 Mega Gene Cluster. *Cell Metab* **27**, 1026-1039 e1026, doi:10.1016/j.cmet.2018.02.022 (2018).
- 11 Sin, J. *et al.* Mitophagy is required for mitochondrial biogenesis and myogenic differentiation of C2C12 myoblasts. *Autophagy* **12**, 369-380, doi:10.1080/15548627.2015.1115172 (2016).
- 12 Chabi, B., Hennani, H., Cortade, F. & Wrutniak-Cabello, C. Characterization of mitochondrial respiratory complexes involved in the regulation of myoblast differentiation. *Cell Biol Int* **45**, 1676-1684, doi:10.1002/cbin.11602 (2021).
- 13 Salucci, S., Baldassarri, V., Falcieri, E. & Burattini, S. alpha-Actinin involvement in Z-disk assembly during skeletal muscle C2C12 cells in vitro differentiation. *Micron* **68**, 47-53, doi:10.1016/j.micron.2014.08.010 (2015).

3. Description of the revisions that have already been incorporated in the transferred manuscript

Reviewer#1

Comment 1, In the first experiments, IPS cells are induced for myogenic conversion by expression of MyoD1. To convincingly demonstrate the conversion of IPS cells, IPS cells prior Dox stimulation should be included in the analysis figure S1/2 to demonstrate an increase in Myogenin expression. Increased expression of myogenin upon MYOD treatment should be demonstrated by Western blot or RT-qPCR. Additional markers of myogenic differentiation should be used before and after induction to demonstrate myogenic differentiation of the IPS cells. If the model discussed later is correct, converted IPS cells should express miR-1 and miR-206 after myogenic conversion and this expression should be dependent on SMN1.

Response : As the reviewer suggested, we examined the sequential expression level of myogenin and MEF2C before and after differentiation (Figure S1C, lines 114-116). As expected, we confirmed that the expression of the genes was negligible before differentiation but upregulated after.

Figure S1C. qPCR analysis for the sequential expression of MyoG and MEF2C.

Comment 2, Loss of SMN1 severely affects cell density in IPS cells, according to the authors due to massive increase of apoptosis. Nevertheless, these cells are used to compare oxygen consumption. In the respective figures it should be made clear how the data were normalized to correct for cell density. In addition, the authors need to exclude that the decreased cell density/dying cells does not impair the measurements. The authors may consider inhibition of apoptosis.

Response: In this study, in order to exclude the possibility that differences in cell number during differentiation affect oxygen consumption, we re-plated a total of 1.0×10^4 iPSCs-derived myogenic cells 5 days after differentiation onto XF96 Cell Culture Microplates and measured oxygen consumption after confirming the cell adhesion. Therefore, Figures 1E and 1F showed the oxygen consumption rate [pmol/min] per approximately 1.0×10^4 cells. We apologize for the confusion caused by inaccuracies in the description. In the legend, we specified that the oxygen consumption rate represents an absolute value per 1.0×10^4 cells.

Comment 3, Similarly, siSmn seems to increase apoptosis of C2C12 cells (figure 1O) and that most likely will affect cell density of the C2C12 cells; cell density affects myotube

Revision Plan

differentiation and it is unclear whether the reduced mitochondrial differentiation is an unspecific effect resulting from impaired myotube formation may be due to lower cell density. Likewise, reduced cell density and thus reduced differentiation could cause the changes in expression of MyoD1, and miRNA expression summarized in figure 2.

Response : As the reviewer pointed out, cell density is important for myogenic differentiation, while mitochondrial maturation promotes myogenic differentiation⁸⁻¹⁰. Moreover, the inhibition of mitochondrial maturation represses the proliferation and differentiation of C2C12, and mitochondrial maturation during initial myogenic differentiation is regulated by MyoD1, miR-1 and -206^{7,9,11,12}. In Figures 3H and 3I, we showed that SMN-depleted C212 treated with miR-1 and -206 improved the cell number and myotube formation. Therefore, we think that the downregulation of MyoD1 and miRs are not just the result of low cell density. It is difficult to completely exclude the effect of the reduced cell density on the downregulation of miRs, because cell proliferation and the upregulation of miRs occur in parallel during initial myogenic differentiation. However, similar results were obtained in multiple cell types. In addition, as shown in the iPS cell model, differences in mitochondrial function were observed even when the cells were reseeded with the same number of cells. Therefore, we believe that SMN affects mitochondria-dependent cell differentiation to some extent.

****Minor comments****

Comment 1, Expression of COX1 protein in C2C12 cells at day 6 (Figure S1L) should be quantified.

Response : We quantified the expression of COX1 protein in C2C12 cells at day 6 (Figures S1N and S1O).

[Figure removed by LSA Editorial Staff per authors' request.]

Revision Plan

Comment 3, The quantification of Western blot data in Fig 6D is missing.

Response : We quantified the expression of mitochondrial protein in figure 6D.

Figure 6.

(D, E) (D) Immunoblotting assay with TA and GA samples from mice (P3) and (E) its quantification.

Reviewer#2

Comment 1, In Fig. 1, 2 and 3. when knocking down SMN with siRNA oligos in C2C12 cells, more than one siRNA should be used. Other methods such as AID induced SMN degradation will be helpful to make the conclusion more solid.

Response : We evaluated the effect of several siSmn oligonucleotide constructs on the downregulation of Smn in C2C12 and used the siSmn showing the highest knockdown efficiency (Figure B, only for the reviewers).

Figure B, only for reviewers. qPCR analysis of Smn1 in C2C12 cells at day 6. Each siSmn has a different target sequence for Smn1 mRNA. The expression level of Smn1 is shown relative to scramble. Rpl13a was used as the internal control. The

error bars indicate means \pm SD. Each dot represents a biologically independent sample. We used siSmn-1 for the subsequent experiments.

Comment 2, More experimental details should be given in the Results. For example, in Figure 1A, how is knockdown and overexpression of SMN achieved to generate related cell lines (B7-MSMNKD and SMA-MOE)?

Response : We apologize for the insufficient details. We added a schema about the establishment of isogenic iPSC lines in the subsection “Human iPSC lines” in the Methods (line 552).

Figure 1.

(A) A schema of the iPSC clones used in this study and the doxycycline-inducible MYOD1-driven myogenic conversion system.

Comment 3, In Figure S2B and S2C, how did the authors distinguish exogenous and endogenous MyoD expression by RT-PCR?

Response : The primer for endogenous MyoD1 was set for the endogenous region of the 3'UTR of the MyoD1 locus, which is not included in the transgene. Since the transgenic MyoD1 is transcribed with mCherry (TetO-MyoD1-IRES-mCherry), we put the primer pair on the mCherry sequence. We renamed the primer pairs to clearly describe the type of transcripts (listed in Table S1).

Comment 6, In Figure 7, to prove miRs treatment can improve satellite cell differentiation, IF staining of MyHC in differentiated satellite cell isolated should be performed.

Response : We performed the muscle differentiation of isolated MuSCs derived from Δ 7-SMA mice and performed IF staining with α ACTN2, a protein associated with sarcomere formation¹³, instead of MyHC. Here, α ACTN2 is an indicator of muscle differentiation as well as MyHC, so it can be a substitute for MyHC. From the results, we concluded improved differentiation by miR treatment.

Minor comments

Comment 1, In Figure 1C, the representative flow cytometry images of caspase-3 expression should be shown, along with the quantification result.

Revision Plan

Response : We added representative flow cytograms of caspase-3 expression

[Figure removed by LSA Editorial Staff per authors' request.]

4. Description of analyses that authors prefer not to carry out

None.

March 21, 2022

Re: Life Science Alliance manuscript #LSA-2022-01457-T

Dr. Megumu K Saito
Kyoto University
Japan

Dear Dr. Saito,

Thank you for submitting your manuscript entitled "SMN promotes mitochondrial metabolic maturation during myogenesis by regulating the MYOD-miRNA axis." to Life Science Alliance. We invite you to re-submit the manuscript, revised according to your Revision Plan.

When submitting the revision, please include a letter addressing the Reviewers' comments point by point.

Thank you for this interesting contribution to Life Science Alliance. We are looking forward to receiving your revised manuscript.

Sincerely,

B. MANUSCRIPT ORGANIZATION AND FORMATTING:

1. General Statements [optional]

We appreciate the reviewers' fruitful comments and suggestions.

2. Description of the planned revisions

Reviewer#1

Comment 4, There is a massive increase in the detection of miRNA host gene RNA between day 3 and day 6, however there is almost no change in the expression of the mature miRNAs. This indicates (beside potential technical shortcomings), that there are massive differences in the processing of the precursor RNAs, thus detection of the host genes or of mature miRNAs, in contrast to the conclusions by the authors, might not reflect regulation of transcriptional activity by a transcription factor. A more direct assay should be used to determine transcriptional activity of the respective genomic loci.

Response: Since the dysregulation of genes associated with miR processing has been previously reported in SMA motor neurons¹, it is also possible that the post-transcriptional dysregulation of miRs simultaneously takes place, as mentioned by the reviewer. Accordingly, we have amended our description (lines 208-209). However, our study focuses on the behavior of SMN on the genome and the transcription of the miR host gene, not the post-transcriptional regulation. It seems clear from the qPCR data that the transcription of the host gene is indeed affected. However, as proposed by the reviewer, it would be better to show the direct effect of SMN on transcription. Therefore, we performed a reporter assay to examine the direct effect of SMN on promoter activity. To this end, we evaluated reporter activity at the previously reported distal and proximal promoter regions of linc-MD1⁶ using C2C12 cells. The experimental schema and results of the reporter assay are shown in Figures 5H and 5I (new Figures). In line with the results of ChIP-qPCR with SMN for the linc-MD1 distal and proximal promoter regions (Figure 5C), the depletion of SMN did not downregulate the activity of the distal promoter (Figure 5I), while the activity of proximal promoter was downregulated in SMN-depleted C2C12 cells (Figure 5I). Thus, SMN indeed binds to the promoter region of Linc-MD1 and regulates transcription.

Comment 5, The maximal oxygen consumption rate depicted in Figure 1 and 3 B/C uses the unit [pmol/min], however the oxygen consumption should be normalized to a defined number of cells or protein content, whatsoever. What is actually the meaning of the additional Scramble=100 in Figure 3B/C? For C2 cells in figure 3B/C the maximal oxygen

consumption rate seems to be quite similar to the spare respiratory capacity. Looking at the Oxygen consumption blots, this appears quite unlikely. Depiction of the respective rates, normalized it cell number is needed.

Response: We apologize for the insufficient description. For the oxygen consumption rate experiments using iPSC -derived skeletal muscle cells shown in Figures 1E and 1F, we normalized the cell number by reseeding 1.0×10^4 cells in each condition into XF96 cell culture microplates on the day before analysis and measured oxygen consumption on the following day. We have modified the corresponding Figure legends and have described this procedure in the Methods section (lines 774-784).

For the oxygen consumption measurement experiments using C2C12 cells (Figures 1K, 1L and Figures 3B, 3C), unlike iPSC-derived myogenic cells, the cells tended to fuse strongly and could therefore not be reseeded immediately before the OCR assay. Therefore, we recovered and counted the cells after oxygen consumption analysis and normalized the oxygen consumption data per 1.0×10^4 cells. The units in the Figures 1K, 1L and Figures 3B, 3C (new Figures) have been corrected to [pmol/min/10000 cells]. We have also added a description for the normalization procedure to the Methods section (lines 782-784).

For the newly added experiments to measure oxygen consumption using skeletal muscle stem cells from SMA model mice (Figures 7D and 7E), as in the experiment with C2C12 cells, the number of cells was measured after analysis and oxygen consumption per 1.0×10^4 cells was shown. This normalization procedure has also been described in the Methods section (lines 782-784).

Regarding the relatively small difference between maximal and spare respiration capacity, we think this is physiologically consistent in cells with high mitochondria activity, such as skeletal muscle cells. The spare respiratory capacity is the maximal respiratory capacity minus the basal respiratory capacity, and the spare respiratory capacity is enhanced with mitochondrial activation. In other words, the spare respiratory capacity reflects the degree of mitochondrial activity (Hong et al., 2020; Yamamoto et al., 2016). In cells with mitochondrial inactivation, such as stem cells, there is almost no difference between basal and maximal respiration capacity and spare respiration capacity are significantly lower than maximal respiration capacity (Sala et al., 2019). On the other hand, in cells with activated mitochondria, the spare respiratory capacity is significantly increased and the maximal respiratory capacity is much higher than the basal respiratory value. Thus, the difference between the maximal and spare respiratory capacity is expected to be relatively smaller in cells with mitochondrial activation, such as skeletal muscle cells.

Comment 6, CHIP-PCR experiments were performed to demonstrate binding of SMN to the promoter of MyoD1 and the miR-1 genes to suggest autoregulation of MyoD1 by MYOD1/SMN and regulation of miR-1 genes by MYOD1/SMN. In addition, Co-IPs were performed to demonstrate interaction of SMN1 with MYOD1. To exclude SMN1 might just interact with RNAPL II and thereby affect transcription of multiple genes, CHIP was

performed targeting ID3 and Rpl11 promoters and no binding of SMN1 was found for those promoters although SMN interaction with RNAPL II was previously described. This is an interesting and important experiment, however not convincing with the limited number of assays. CHIP-seq of MYOD1, SMN1 and RNAPL II is essential to demonstrate the overlap in DNA binding by MYOD1 and SMN in an unbiased manner, as well as to understand the significance of the SMN - RNAPL II interaction in this model.

Response: We would like to thank the reviewer for recognizing the significance of our CHIP-PCR experiments. As pointed out, it is important to show the binding sites where SMN and MYOD1 co-localize throughout the genome by ChIP-seq. We originally planned to use the ChIP-seq dataset for MYOD1 using C2C12 cells (GEO accession code: GSM915186), as suggested by reviewer 2 (comment 4). However, we failed to perform the ChIP-seq of SMN with C2C12, and found that the quality of the dataset was insufficient. Therefore, we performed a series of ChIP-seq using human iPSC-derived skeletal muscle cells. For the ChIP-seq of MYOD1, we generated a new human iPSC line introduced with doxycycline-inducible FLAG-MYOD1, and examined the binding sites of MYOD1 using anti-FLAG antibody. The methods have been described in the Methods section (lines 552-555, 754-767).

We first classified the genes into clusters 1, 2, and 3 according to the binding manners of RNAPLII (Figure S5F). In cluster 2, which represents a group of genes undergoing active transcription, not only RNAPLII and MYOD1 but also SMN showed a high binding around TSS (Figure S5G). Further analysis of the genome-wide binding regions of RNAPLII, MYOD1, and SMN showed co-localization at specific loci, including the promoter region of linc-MD1 (Figure S5H, S5I). The binding of SMN was absent at some MYOD1-bound loci, such as MYOG, MEF2C, and MRF4, indicating that SMN is not essential for the regulation of all genes by MYOD1 (lines 332-334). The genomic binding sites of SMN and RNAPLII did not completely overlap, suggesting that SMN is not completely subordinated to RNAPLII (Figure S5I).

In ChIP-qPCR using C2C12 cells, no binding of SMN to the promoter region of Id3 was observed. However, in ChIP-seq using human iPSC-derived skeletal muscle cells, the binding of SMN to the ID3 locus was observed. This may be due to species differences. Therefore, we searched for a common gene among species that is regulated by MYOD1 but does not bind SMN, and found that the MYOG1 gene fulfills such a condition. Therefore, the ChIP-PCR data for Id3 has been replaced with that of Myog in the revised paper (Figure 5E).

In summary, SMN cooperates with RNAPLII and binds to the TSS region of certain MYOD1-regulated genes. Combined with the results of the reporter assay (Figures 5H and 5I), SMN is likely to be specifically involved in the transcriptional regulation of these genes.

Comment 7, To strengthen the findings obtained in C2C12 cells, the authors also analyzed the mitochondrial area in muscle of delta7-SMA mice. Unfortunately, this part of the

manuscript is particularly weak. It is not surprising that mice already suffering atrophy of the muscles also reveal changes in mitochondrial content of the muscle. A muscle-specific deletion/modification of SMN would be needed to address the muscle specific function of SMN1.

Response: As pointed out by the reviewer, it is a natural consequence that the amount of mitochondria in skeletal muscle decreases with muscle atrophy. However, we found that, in SMA mice, mitochondrial dysregulation takes place prior to the motor neuron denervation, which was not described in previous reports^{2,3}. We also identified novel aberrant functions in skeletal muscle stem cells, as discussed below.

We focused on skeletal muscle stem cells isolated from SMA model mice to analyze their functions in an *ex vivo* culture system. Therefore, we believe that skeletal muscle-specific SMN knockout mice are not necessary. However, as pointed out by the reviewer, we need to demonstrate functional abnormalities in skeletal muscle stem cells derived from SMA model mice more clearly. Thus, we examined the oxygen consumption capacity and expression of MyoD1, miR-1, and miR-206 and their host genes using isolated skeletal muscle stem cells. The expression of MyoD1, miR-1, and miR-206 and their host genes was found to be downregulated during the differentiation of skeletal muscle stem cells (Figures 6F-6J). Accordingly, maximal oxygen consumption capacity was also decreased in myotube cells differentiated from skeletal muscle stem cells derived from SMA model mice (Figures 7D). These results are in principle consistent with the results using C2C12 cells and iPS cells. Since the effects of other cell types, such as neurons, are excluded in these experimental systems, the data suggest that even, in primary mouse muscle stem cells, the depletion of SMN causes cell-autonomous aberrant functional consequences.

Comment 8, Unfortunately, the analysis of MuSCs isolated from delta7-SMA animals at P5 also is superficial and does not relate to the mechanism suggest in the manuscript. While the authors include data on the contractility of myotubes differentiated from the MuSCs, data on MyoD1 and miRNA expression are missing, no data on the differentiation ability of MuSCs and mitochondrial activity are included. Thus, the miRNA supplementation assay remains incomplete and inconclusive without value to support the model figure 8.

Response: In response to the reviewer's comment, additional experiments were performed to verify whether the data from skeletal muscle stem cells derived from SMA model mice supported the model shown in Figure 8.

As discussed in comment 7, similar to the results for C2C12 cells and iPSC-derived skeletal muscle cells, SMA model mouse-derived skeletal muscle stem cells showed decreased expression of MyoD1, miR-1, and miR-206 and their host genes during differentiation (Figures 6F-6J). Furthermore, myotube cells differentiated from skeletal muscle stem cells derived from SMA model mice showed reduced oxygen consumption compared to those from wild-type mice. The reduction in oxygen consumption was improved by transfection with miR-1 and miR-206, supporting the responsibility of the miR-

1 and miR-206 axes to impaired myogenesis of skeletal muscle stem cells derived from SMA model mice (Figures 7D, 7E). Regarding differentiation ability, we have already shown that the introduction of miR-1 and miR-206 improved the myotube cell formation ability of skeletal muscle stem cells derived from SMA model mice, which had been reduced (Figures 7J, 7K, S7E and S7F).

****Minor comments****

Comment 2, SMN is upregulated in IPS cells overexpressing Myod1 and in differentiating C2C12 cells. This goes along with increasing amounts of protein in the nucleus. This results do not justify the statement that SMN during myogenesis might have a unique role in the nucleus, because SMN seems to be in the nucleus also prior to differentiation. The figure 4f also should contain total lysate day 0.

Response: We did not mean to conclude that SMN has a unique role in the nucleus during myogenic differentiation from only our observations in Figures 4D and S4C. However, we noticed that SMN has a characteristic nuclear distribution during differentiation, which is different from typical Cajal body localization, and constructed a hypothesis accordingly. Therefore, we have amended the text to reduce the strength of our conclusions (line 246). Following the reviewer's suggestion, total lysate at day 0 was added to Figure 4F, which clarifies the relative increase in nuclear SMN during myogenesis.

Figure A, only for the reviewers. Immunostaining of iPSC-derived myogenic cells on

day 3 with SMN.

Green indicates SMN. The left-hand image shows typical localization in undifferentiated iPSCs. The right-hand image shows the transient, diffuse localization observed during myogenesis. White arrows indicate the typical localization of SMN in the nuclei.

Reviewer#2

Comment 3, From the experiments in figure 3 it remains unclear whether the decrease in miRNAs directly affects mitochondrial maturation or is a consequence of reduced differentiation of myoblasts to myotubes.

Response: Although we are also interested in this point, it is difficult to clearly distinguish the effect of miRNAs on mitochondrial function and mitochondria-independent myogenesis. In the initial myogenic differentiation, it is known that miR-1 and miR-206 have another target to promote differentiation except for mitochondria, and linc-MD1 also works as competing endogenous RNA in the cytoplasm.⁴⁻⁶ Therefore, the downregulation of SMN could impair both cytoplasmic and mitochondrial pathways during initial myogenic differentiation. Although mitochondrial maturation is an important event, coordinated differentiation is essential to promote differentiation^{4,5,7}. In this study, we focused on the MyoD1-miR-mitochondrial maturation pathway. However, the contribution of other pathways to SMN muscle maturation will be investigated in future studies. We have included these points in the Discussion of the revised manuscript (lines 419-428).

Comment 4, In Figure 5, SMN ChIP-seq can be performed in C2C12 cells; co-binding of SMN and MyoD can be demonstrated by integrating with existing MyoD ChIP-seq dataset in C2C12 (GEO accession code:GSM915186).

Response: As both reviewers have pointed out, it is important to show the binding sites where SMN and MYOD1 co-localize throughout the genome by ChIP-seq. We originally intended to use the ChIP-seq dataset for MYOD1 using C2C12 cells (GEO accession code: GSM915186), as suggested by reviewer 2 (comment 4). However, we failed to perform the ChIP-seq of SMN with C2C12, and found that the quality of the dataset was insufficient. Therefore, we performed a series of ChIP-seq using human iPSC-derived skeletal muscle cells. For ChIP-seq of MYOD1, we generated a new human iPSC line introduced with doxycycline-inducible FLAG-MYOD1, and examined the binding sites of MYOD1 using anti-FLAG antibody. The methods used have been described in the Methods section (lines 545-548, 747-760).

We first classified the genes into clusters 1, 2, and 3 according to the binding manners of RNAPLII (Figure S5F). In cluster 2, which represents a group of genes undergoing active transcription, not only RNAPLII and MYOD1 but also SMN showed a high binding around TSS (Figure S5G). Further analysis of the genome-wide binding regions of RNAPLII, MYOD1, and SMN showed co-localization at specific loci, including the promoter region of linc-MD1 (Figure S5H, S5I). The binding of SMN was absent at some MYOD1-bound loci, such as MYOG, MEF2C, and MRF4, indicating that SMN is not essential for the regulation of all genes by MYOD1 (lines 332-334). The genomic binding sites of SMN and RNAPLII did not completely overlap, suggesting that SMN is not completely subordinated to RNAPLII (Figure S5I).

In summary, SMN cooperates with RNAPLII and binds to the TSS region of certain MYOD1-regulated genes. Combined with the results of the reporter assay (Figures 5H and 5I), SMN is likely to be specifically involved in the transcriptional regulation of these genes.

Comment 5, In Figure 6, to show the mitochondrial dysfunction in the skeletal muscle of SMA model mice, muscle stem cells from WT and $\Delta 7$ -SMA mice should be isolated to measure the mitochondrial oxygen consumption rate as done in Figure 1E.

Response: According to the reviewer's suggestion, we isolated CD45⁻ CD31⁻ Sca⁻¹ Integrin $\alpha 7$ + skeletal muscle stem cells from wild-type and SMA model mice (Figure S6J). The oxygen consumption rate was measured after myogenic differentiation from the skeletal muscle stem cells. As a result, we confirmed that the myotubular cells from SMA model mice had a lower oxygen consumption than those from wild-type mice (Figure 7D, 7E).

****Minor comments****

Comment 2, In experiments overexpressing miR-1 and miR-206, the overexpression efficiency should be shown.

Response: Transduction efficiency into isolated skeletal muscle stem cells and the expression levels of miR-1 and miR-206 after transduction were examined. The transduction efficiency of the expression vectors measured by the fluorescent signal of GFP incorporated in the expression vectors showed a very high efficiency (Figure S7A). In line with this, the expression levels of miR-1 and miR-206 increased after transduction (Figure S7B).

3. Description of the revisions that have already been incorporated in the transferred manuscript

Reviewer#1

Comment 1, In the first experiments, IPS cells are induced for myogenic conversion by expression of MyoD1. To convincingly demonstrate the conversion of IPS cells, IPS cells prior Dox stimulation should be included in the analysis figure S1/2 to demonstrate an increase in Myogenin expression. Increased expression of myogenin upon MYOD treatment should be demonstrated by Western blot or RT-qPCR. Additional markers of myogenic differentiation should be used before and after induction to demonstrate myogenic differentiation of the IPS cells. If the model discussed later is correct, converted IPS cells should express miR-1 and miR-206 after myogenic conversion and this expression should be dependent on SMN1.

Response: As the reviewer suggested, we examined the sequential expression level of myogenin and MEF2C before and after differentiation (Figure S1C, lines 113-114). As expected, we confirmed that the expression of the genes was negligible before differentiation but upregulated after that.

Figure S1C. qPCR analysis for the sequential expression of MyoG and MEF2C.

Comment 2, Loss of SMN1 severely affects cell density in IPS cells, according to the authors due to massive increase of apoptosis. Nevertheless, these cells are used to compare oxygen consumption. In the respective figures it should be made clear how the data were normalized to correct for cell density. In addition, the authors need to exclude that the decreased cell density/dying cells does not impair the measurements. The authors may consider inhibition of apoptosis.

Response: In this study, in order to exclude the possibility that differences in cell number during differentiation affect oxygen consumption, we re-plated a total of 1.0×10^4 iPSCs-derived myogenic cells 5 days after differentiation onto XF96 Cell Culture Microplates and measured oxygen consumption after confirming the cell adhesion. Therefore, Figures 1E and 1F showed the oxygen consumption rate [pmol/min] per 1.0×10^4 cells. We apologize for the confusion caused by inaccuracies in the description. In the legend, we specified that the oxygen consumption rate represents an absolute value per 1.0×10^4 cells.

Comment 3, Similarly, siSmn seems to increase apoptosis of C2C12 cells (figure 1O) and that most likely will affect cell density of the C2C12 cells; cell density affects myotube differentiation and it is unclear whether the reduced mitochondrial differentiation is an unspecific effect resulting from impaired myotube formation may be due to lower cell density. Likewise, reduced cell density and thus reduced differentiation could cause the changes in expression of MyoD1, and miRNA expression summarized in figure 2.

Response: As the reviewer pointed out, cell density is important for myogenic differentiation, while mitochondrial maturation promotes myogenic differentiation⁸⁻¹⁰. Moreover, the inhibition of mitochondrial maturation represses the proliferation and differentiation of C2C12, and mitochondrial maturation during initial myogenic differentiation is regulated by MyoD1, miR-1 and -206^{7,9,11,12}. In Figures 3H and 3I, we showed that SMN-depleted C212 treated with miR-1 and -206 improved the cell number and myotube formation. Therefore, we think that the downregulation of MyoD1 and miRs are not just the result of low cell density. It is difficult to completely exclude the effect of the reduced cell density on the downregulation of miRs, because cell proliferation and the upregulation of miRs occur in parallel during initial myogenic differentiation. However, similar results were obtained in multiple cell types. In addition, as shown in the iPSC cell model, differences in mitochondrial function were observed even when the cells were reseeded with the same number of cells. Therefore, we believe that SMN affects mitochondria-dependent cell differentiation to some extent.

Minor comments

Comment 1, Expression of COX1 protein in C2C12 cells at day 6 (Figure S1L) should be quantified.

Response : We quantified the expression of COX1 protein in C2C12 cells at day 6 (Figures S1N and S1O).

[Figure removed by LSA Editorial Staff per authors' request.]

Comment 3, The quantification of Western blot data in Fig 6D is missing.

Response : We quantified the expression of mitochondrial protein in figure 6D. Figure 6.

[Figure removed by LSA Editorial Staff per authors' request.]

Reviewer#2

Comment 1, In Fig. 1, 2 and 3. when knocking down SMN with siRNA oligos in C2C12 cells, more than one siRNA should be used. Other methods such as AID induced SMN degradation will be helpful to make the conclusion more solid.

Response : We evaluated the effect of several siSmn oligonucleotide constructs on the downregulation of Smn in C2C12 and used the siSmn showing the highest knockdown efficiency (Figure B, only for the reviewers).

Figure B, only for reviewers. qPCR analysis of Smn1 in C2C12 cells at day 6. Each siSmn has a different target sequence for Smn1 mRNA. The expression level of Smn1 is shown relative to scramble. Rpl13a was used as the internal control. The error bars indicate means \pm SD. Each dot represents a biologically independent sample. We used siSmn-1 for the subsequent experiments.

Comment 2, More experimental details should be given in the Results. For example, in Figure 1A, how is knockdown and overexpression of SMN achieved to generate related cell lines (B7-MSMNKD and SMA-MOE)?

Response : We apologize for the insufficient details. We added a schema about the establishment of isogenic iPSC lines in the subsection “Human iPSC lines” in the Methods (line 635).

Figure 1.
(A) A schema of the iPSC clones used in this study and the doxycycline-inducible MYOD1-driven myogenic conversion system.

Comment 3, In Figure S2B and S2C, how did the authors distinguish exogenous and endogenous MyoD expression by RT-PCR?

Response : The primer for endogenous MyoD1 was set for the endogenous region of the 3'UTR of the MyoD1 locus, which is not included in the transgene. Since the transgenic MyoD1 is transcribed with mCherry (TetO-MyoD1-IRES-mCherry), we put the primer pair on the mCherry sequence. We renamed the primer pairs to clearly describe the type of transcripts (listed in Table S1).

Comment 6, In Figure 7, to prove miRs treatment can improve satellite cell differentiation, IF staining of MyHC in differentiated satellite cell isolated should be performed.

Response : In fact, we performed muscle differentiation of isolated MuSCs derived from $\Delta 7$ -SMA mice and performed IF staining with α ACTN2, a protein associated with sarcomere formation¹⁴, instead of MyHC. Here, α ACTN2 staining is the same meaning with MyHC, and we believe that it indicates improved differentiation by miR treatment.

****Minor comments****

Comment 1, In Figure 1C, the representative flow cytometry images of caspase-3 expression should be shown, along with the quantification result.

Response : We added representative flow cytograms of caspase-3 expression

[Figure removed by LSA Editorial Staff per authors' request.]

4. Description of analyses that authors prefer not to carry out

None.

- 1 Goncalves, I. *et al.* Neuronal activity regulates DROSHA via autophagy in spinal muscular atrophy. *Sci Rep* **8**, 7907, doi:10.1038/s41598-018-26347-y (2018).
- 2 Ripolone, M. *et al.* Impaired Muscle Mitochondrial Biogenesis and Myogenesis in Spinal Muscular Atrophy. *JAMA Neurol* **72**, 666-675, doi:10.1001/jamaneurol.2015.0178 (2015).
- 3 Kim, J. K. *et al.* Muscle-specific SMN reduction reveals motor neuron-independent disease in spinal muscular atrophy models. *J Clin Invest* **130**, 1271-1287, doi:10.1172/JCI1131989 (2020).
- 4 Chen, J. F. *et al.* The role of microRNA-1 and microRNA-133 in skeletal muscle proliferation and differentiation. *Nat Genet* **38**, 228-233, doi:10.1038/ng1725 (2006).
- 5 Chen, J. F. *et al.* microRNA-1 and microRNA-206 regulate skeletal muscle satellite cell proliferation and differentiation by repressing Pax7. *J Cell Biol* **190**, 867-879, doi:10.1083/jcb.200911036 (2010).
- 6 Cesana, M. *et al.* A long noncoding RNA controls muscle differentiation by functioning as a competing endogenous RNA. *Cell* **147**, 358-369, doi:10.1016/j.cell.2011.09.028 (2011).
- 7 Shintaku, J. *et al.* MyoD Regulates Skeletal Muscle Oxidative Metabolism Cooperatively with Alternative NF-kappaB. *Cell Rep* **17**, 514-526, doi:10.1016/j.celrep.2016.09.010 (2016).
- 8 Messina, G. *et al.* p27Kip1 acts downstream of N-cadherin-mediated cell adhesion to promote myogenesis beyond cell cycle regulation. *Mol Biol Cell* **16**, 1469-1480, doi:10.1091/mbc.e04-07-0612 (2005).
- 9 Zhang, X. *et al.* MicroRNA directly enhances mitochondrial translation during muscle differentiation. *Cell* **158**, 607-619, doi:10.1016/j.cell.2014.05.047 (2014).
- 10 Wust, S. *et al.* Metabolic Maturation during Muscle Stem Cell Differentiation Is Achieved by miR-1/133a-Mediated Inhibition of the Dlk1-Dio3 Mega Gene Cluster. *Cell Metab* **27**, 1026-1039 e1026, doi:10.1016/j.cmet.2018.02.022 (2018).
- 11 Sin, J. *et al.* Mitophagy is required for mitochondrial biogenesis and myogenic differentiation of C2C12 myoblasts. *Autophagy* **12**, 369-380, doi:10.1080/15548627.2015.1115172 (2016).
- 12 Chabi, B., Hennani, H., Cortade, F. & Wrutniak-Cabello, C. Characterization of mitochondrial respiratory complexes involved in the regulation of myoblast differentiation. *Cell Biol Int* **45**, 1676-1684, doi:10.1002/cbin.11602 (2021).
- 13 Ieronimakis, N. *et al.* Absence of CD34 on murine skeletal muscle satellite cells marks a reversible state of activation during acute injury. *PLoS One* **5**, e10920, doi:10.1371/journal.pone.0010920 (2010).
- 14 Salucci, S., Baldassarri, V., Falcieri, E. & Burattini, S. alpha-Actinin involvement in Z-disk assembly during skeletal muscle C2C12 cells in vitro differentiation. *Micron* **68**, 47-53, doi:10.1016/j.micron.2014.08.010 (2015).

November 25, 2022

RE: Life Science Alliance Manuscript #LSA-2022-01457-TR

Dr. Megumu K Saito
Kyoto University
53, Shogoin-Kawahara, Sakyo
Kyoto 6068507
Japan

Dear Dr. Saito,

Thank you for submitting your revised manuscript entitled "SMN promotes mitochondrial metabolic maturation during myogenesis by regulating the MYOD-miRNA axis.". We would be happy to publish your paper in Life Science Alliance pending final revisions necessary to meet our formatting guidelines.

- please address the final Reviewer 1's minor comments
- please upload your Graphical Abstract as a separate file designated as a Graphical Abstract
- please consult our manuscript preparation guidelines <https://www.life-science-alliance.org/manuscript-prep> and make sure your manuscript sections are in the correct order
- please upload your supplementary figures as single files and add the supplementary figure legends, video legends, and table legends to the main manuscript text
- please add the Twitter handle of your host institute/organization as well as your own or/and one of the authors in our system

Figure Check:

- Figure 7L, Figure S6C and Figure S7G need scale bars
- Figure S6H: looks like there is a splice in top and bottom row between the blots. Please provide source data

A. FINAL FILES:

B. MANUSCRIPT ORGANIZATION AND FORMATTING:

Sincerely,

Reviewer #1 (Comments to the Authors (Required)):

The authors have addressed my concerns in the revision. I have some minor comments for the new version.

1. The defect of differentiation in Figure 1H is obvious, but the quantification in Figure 1I using the ratio of DAPI-positive area to MyHC-positive area is uncommon. The number of MyHC+ fibers per field or the average number of nuclei per fiber is more commonly used.
2. In Figure 4D, merged images of DAPI and SMN should be presented to show the distribution of SMN in the nuclei.
3. In line 294-295 and Figure 5G, the authors states, "As expected, the expression of these genes was not significantly different between C2C12siSmn and C2C12sc (Figure 5G)". Considering MyoG is a direct target of MyoD and a marker of differentiation, its expression should be repressed in siSmn cells since MyoD expression was obviously decreased and C2C12 differentiation was impaired in C2C12siSmn cells. Moreover, the p (0.002) value for Rpl11 should be double checked since the expression of Rpl11 should not show significant difference in Figure 5G.

COMMENT FROM THE EDITOR**Comment**

Figure 7L, Figure S6C and Figure S7G need scale bars

Response

We added scale bars in Figure S6C, 7L and S7G.

Comment

Figure S6H: looks like there is a splice in top and bottom row between the blots. Please provide source data

Image A : SMN (38 kDa) Image B : COX1 (35 kDa) Image C : β -Actin (45 kDa)

Response

The uncropped gel images of Figure S6H is shown above. The regions surrounded by red square are shown in Figure S6H. These images have been uploaded as source data.

COMMENT FROM REVIEWER #1**Comment by reviewer#1**

1. The defect of differentiation in Figure 1H is obvious, but the quantification in Figure 1I using the ratio of DAPI-positive area to MyHC-positive area is uncommon. The number of MyHC⁺ fibers per field or the average number of nuclei per fiber is more commonly used.

Response

As proposed by the reviewer, we quantified the number of MyHC⁺ fibers per field in Figure 1H and revised the figure and legends.

Comment

2. In Figure 4D, merged images of DAPI and SMN should be presented to show the distribution of SMN in the nuclei.

Response

As proposed by the reviewer, we show the merged images of DAPI and SMN in Figure 4D.

Comment

3. In line 294-295 and Figure 5G, the authors states, "As expected,the expression of these genes was not significantly different between C2C12siSmn and C2C12sc (Figure 5G)". Considering MyoG is a direct target of MyoD and a marker of differentiation, its expression should be repressed in siSmn cells since MyoD expression was obviously decreased and C2C12 differentiation was impaired in C2C12siSmn cells. Moreover, the p (0.002) value for Rpl11 should be double checked since the expression of Rpl11 should not show significant difference in Figure 5G.

Response

We are very grateful to the reviewer for his/her careful checking of the figures, and we apologize for the incorrect statistical analysis in Figure 5G. Statistical analysis was performed again for the expression of RPL11 in Figure 5G. The correct p value for the analysis was 0.6155, which was considered as not significant. In relation to this point raised by the reviewer, we reviewed all the statistical data and confirmed that there was no error other than Figure 5G.

December 13, 2022

RE: Life Science Alliance Manuscript #LSA-2022-01457-TRR

Dr. Megumu K Saito
Kyoto University
53, Shogoin-Kawahara, Sakyo
Kyoto 6068507
Japan

Dear Dr. Saito,

Thank you for submitting your Research Article entitled "SMN promotes mitochondrial metabolic maturation during myogenesis by regulating the MYOD-miRNA axis.". It is a pleasure to let you know that your manuscript is now accepted for publication in Life Science Alliance. Congratulations on this interesting work.

DISTRIBUTION OF MATERIALS:

Again, congratulations on a very nice paper. I hope you found the review process to be constructive and are pleased with how the manuscript was handled editorially. We look forward to future exciting submissions from your lab.

Sincerely,
